psychology/neuroscience/cognition

Bayesian perception, tool use,
sensorimotor control, causal inference, agency

**Author for correspondence:**
Nienke B. Debats
e-mail: nienke.debats@uni-bielefeld.de

# Exploring the time window for causal inference and the multisensory integration of actions and their visual effects

## Nienke B. Debats[1,2] and Herbert Heuer[1,3]

[1]Department of Cognitive Neuroscience, [2]Cognitive Interaction Technology Center of Excellence (CITEC), Universität Bielefeld, Bielefeld, Germany
[3]Leibniz Research Centre for Working Environment and Human Factors, Dortmund, Germany

NBD, 0000-0001-6941-949X; HH, 0000-0002-0488-2878

Successful computer use requires the operator to link the movement of the cursor to that of his or her hand. Previous studies suggest that the brain establishes this perceptual link through multisensory integration, whereby the causality evidence that drives the integration is provided by the correlated hand and cursor movement trajectories. Here, we explored the temporal window during which this causality evidence is effective. We used a basic cursor-control task, in which participants performed out-and-back reaching movements with their hand on a digitizer tablet. A corresponding cursor movement could be shown on a monitor, yet slightly rotated by an angle that varied from trial to trial. Upon completion of the backward movement, participants judged the endpoint of the outward hand or cursor movement. The mutually biased judgements that typically result reflect the integration of the proprioceptive information on hand endpoint with the visual information on cursor endpoint. We here manipulated the time period during which the cursor was visible, thereby selectively providing causality evidence either before or after sensory information regarding the to-be-judged movement endpoint was available. Specifically, the cursor was visible either during the outward or backward hand movement (conditions *Out* and *Back*, respectively). Our data revealed reduced integration in the condition *Back* compared with the condition *Out*, suggesting that causality evidence available before the to-be-judged movement endpoint is more powerful than later evidence in determining how strongly the brain integrates the endpoint information. This finding further suggests that sensory integration is not delayed until a judgement is requested.

# 1. Background

During everyday computer use, humans effortlessly steer a cursor on the computer monitor either by whole-hand movements with a mouse or stylus, or by finger movements on a trackpad. This skilled sensorimotor behaviour implies that the actors know that the moving image of the cursor belongs to the movement of their finger or hand. Or in other words, the brain seems to establish a perceptual link between an action (the hand or finger movement) and its distant visual consequence (the cursor movement). Experimentally, this link was indicated by the observation that visual information regarding a cursor is processed faster than information regarding similar visual images on the monitor [1]. This is only possible when the brain identifies the visual information from the cursor as being related to the proprioceptive information from hand or finger movement. Furthermore, it was repeatedly observed in cursor-control tasks that judgements of the hand position were systematically biased to a—slightly deviating—cursor position, and vice versa for the judgements of the cursor position (e.g. [2–8]). In the current study, we address the time window during which this link between actions and their distant visual consequences is established.

Here, we explore the characteristics of the perceptual link between actions and their visual effects by means of a basic cursor-control paradigm of the same type as in previous studies (e.g. [3,5,8,9]). In this paradigm, participants perform out-and-back reaching movements with their occluded right hand on a horizontal digitizer tablet. The reaching movements start from a central position and end at different positions along a semicircular physical border. The corresponding cursor motion is shown on a monitor placed vertically in front of the participants. However, the cursor motion is experimentally manipulated. Specifically, its direction is rotated relative to the direction of the hand movement in order to create a small discrepancy between the hand and cursor endpoints. After a return movement to the start position, participants report the reach endpoint of the hand or of the cursor. These position judgements typically reveal systematic mutual biases: the hand position judgements are biased towards where the cursor would be if projected onto the same plane of motion, and vice versa for the cursor position judgements. Thus, participants are unable to correctly judge the endpoint of a reaching movement due to seeing a cursor moving in a slightly different direction. Similarly, they are unable to correctly judge the final position of a cursor motion on a monitor due to them moving their hand in a slightly different direction. These systematic biases across the two workspaces are a clear indication that the brain links position information from the hand with that of the cursor. This link is so persistent that the biases exist even after participants are informed about the spatial discrepancies and explicitly instructed not to be distracted by the other modality [10].

In a previous study, we found that the perceptual link between the hand and the cursor obeys the principles of optimal multisensory integration [8]. Multisensory integration tags the process through which the brain combines redundant sensory information, such as visual and haptic information on the size of a handheld object, or auditory and visual information on the location of a person speaking (for reviews, see [11–13]). The integrated perceptual estimate can be described as a weighted average of the unisensory estimates, and for optimal integration the weights scale with the unisensory relative reliabilities (i.e. inverse variance). This kind of integration has been observed both within and across sensory modalities, and for a wide variety of perceptual tasks (e.g. [14–17]). The typical change in weights according to the 'reliability rule' [18] is also what we observed in the cursor-control paradigm. That is, as we manipulated the reliability of the unisensory hand and cursor position estimates, we observed a corresponding shift in the biases [8].

Given that only a subset of all the incoming sensory information is redundant, the process of integrating redundant sensory information needs to be preceded by a process in which sensory redundancies are assessed. This latter process is referred to as causal inference (alternative designations are correspondence problem or unity assumption) (for reviews, see [19–21]). Causal inference can vary in strength. It is conceived as a graded belief that two—or more—signals are redundant. With a highly certain causality judgement, the different sensory signals are treated as referring to the same object feature and sensory information is fully integrated (figure 1a). This full integration is the standard view on optimal integration: the integrated estimate is a weighted average of the unisensory signals whereby the weights sum up to one (e.g. [16]). For a less certain causality judgement, the integration is partial (also referred to as breakdown of integration). In this case, the weights add up to less than one, which can be observed by mutually biased bimodal judgements (figure 1b). Theoretical conceptions of partial integration generally posit gradually increasing biases with gradually increasing certainty of the causality judgement (e.g. [13,22–24]). Integration strength, as defined by the sum of the weights (which, as shown below, correspond to proportional biases),

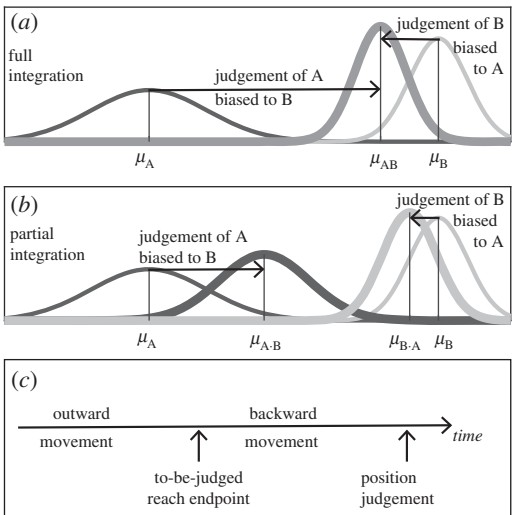

**Figure 1.** Experimental rationale. (*a*) This panel illustrates the integration of unisensory signals A and B (the thin-lined Gaussian distributions) into a bimodal estimate of A-in-the-presence-of-B or B-in-the-presence-of-A (the thick-lined Gaussian distributions). The black arrows indicate the magnitude of the biases for full integration of A and B. (*b*) This panel shows the partial integration of A and B: the bimodal estimates are mutually biased as in (*a*), but biases do not cover the full distance between A and B. (*c*) This panel sketches the timeline of the key ingredients of the experimental task: participants make an out-and-back reaching movement, with a cursor being shown at the endpoint. This endpoint of the hand or cursor is to be judged after the backward return movement. The causality information (available when the cursor movement is shown during the hand movement) was provided either during the outward movement and thus before the to-be-judged endpoint was reached (condition *Out*), or it was provided during the backward movements and thus in between reaching the to-be-judged endpoint and making the judgement (condition *Back*).

thus directly reflects the certainty of the causality judgement. Overall, causal inference and optimal integration can thus be considered as two closely related processes that enable the identification and integration of redundant sensory information, whereby differences in integration strength reveal underlying differences in the certainty of the causality judgements.

There are several potential sources of information on which causal inference can be based: first, causality evidence can be a characteristic of the redundant sensory signals themselves, in particular the coherence of the sensory information in the temporal domain (e.g. audio–visual estimate of the position of speech source in ventriloquism) and in the spatial domain (e.g. visuo-haptic estimate of the size of a handheld object) (e.g. [25–28]). Second, causal inference can be based on cognitive factors such as instructed knowledge of discrepancies between sensory signals [29–31]. Third, causal inference can be influenced by previous sensory experience or by additional sensory information [32–34]. To give an example, the visuo-haptic integration of the seen and felt size of a handheld object breaks down when the visual and haptic size information come from different spatial locations. Yet it is restored based on additional visual information of a virtual tool that covers the distance between these locations [33]. Together, these sources of causality evidence determine the causality judgement.

The integration of hand and cursor position information seems striking, given that the position information here is not truly redundant. That is, the hand and cursor are two distinct 'objects' in separate planes of motion. Nevertheless, the sum of the proportional biases in the hand–cursor position judgements (i.e. the strength of the integration) typically lies around 0.75, thus indicating substantial partial integration [8–10,29,35]. In a series of previous studies, we examined what causality evidence underlies this integration. In one of those studies, we manipulated the cursor movement such that it moved in a different way (i.e. it included an additional curvature) than the hand, a manipulation through which the cross-correlations between kinematic parameters (e.g. instantaneous direction or speed) was reduced [9]. This resulted in a clear decrease in integration strength, indicating that the correlated hand–cursor movement trajectories provide a prominent source of causality evidence. In another study, we found that the relation between the hand and cursor planes of motion (either both horizontal, or one horizontal and the other vertical) did not affect the integration strength [35]. Finally, we examined the integration strength when the cursor was only shown in the movement endpoint [29]. The absence of simultaneous hand and cursor motion means an absence of the

causality evidence from the correlated trajectories. What we observed was a reduced level of integration strength. This indicates the involvement of a causality prior, probably based on the long-term experience of the correlated hand and cursor trajectories during everyday computer work, in addition to the immediate sensory causality evidence available in the correlated hand–cursor movements ending at the hand and cursor positions to be judged.

An interesting aspect of the integration of hand and cursor position information is that the causality evidence (correlated hand–cursor movement trajectories) and the to-be-judged sensory information (static movement endpoints) are available at separate moments in time. This prompted us to further explore the time window for causal inference and optimal integration, in particular: if causality information is presented later than the to-be-judged sensory information, does it promote its integration in a retrospective manner? Or in terms of the cursor-control paradigm: is the hand–cursor integration equally strong with causality information present (i.e. by the cursor being visible) during the outward movement versus during the backward movement? Important here is that position judgements are requested upon completion of the backward movement such that in both cases the causality information is present beforehand (figure 1*c*).

We are not aware of well-founded expectations for the answer to this question. On the one hand, integration might directly follow causal inference in order to facilitate perception with minimal delay. Along this line of reasoning one would expect that the redundant sensory information is integrated immediately based on the causality evidence available up to that time-point. Causality evidence provided thereafter should be ineffective because unisensory estimates are no longer available. On the other hand, the brain could maintain the unisensory estimates and integrate them only once required to judge a position for which redundant information is available, making use of all causality evidence up to that occasion.

In short, we here compared the effectiveness of causality evidence that precedes the to-be-judged sensory information with that of causality evidence that directly follows the to-be-judged sensory information. We used the cursor-control task described above and presented the cursor (and hence the causality evidence) either during the outward hand movement (condition *Out*) or during its backward movement (condition *Back*). We quantified the strength of the integration and used it as a measure for the effectiveness of the underlying causality evidence. To assess potential differences between the condition in terms of quality and quantity of the causality evidence, we analysed the movement trajectories. And to assess potential deviations from the principles of optimal integration, we compared the characteristics of the biased position judgements with model predictions using the same model as in our previous studies of multisensory integration in cursor control [8–10,29,35].

# 2. Methods

## 2.1. Participants

Fourteen right-handed participants (aged 19–26 years; nine females) volunteered to take part in the experiment. All participants were naive with respect to the purpose of the experiment. They gave written informed consent prior to their participation and were compensated with a payment of €6 per hour. The experiment was conducted in accordance with the declaration of Helsinki and approved by the Bielefeld University ethics committee.

## 2.2. Apparatus

The apparatus is illustrated in figure 2*a*. Participants sat at a table with a digitizer tablet on top of it (Wacom Intuos4 XL; 48.8 by 30.5 cm). A chinrest supported them in keeping a 60 cm viewing distance from a computer monitor (Samsung MD230; 23 inches; 50.9 by 28.6 cm). They held a stylus in their right hand and pressed a button on the stylus with their thumb or index finger when required. A semicircular workspace of 15 cm radius was created on the tablet by means of a 5 mm thick PVC template. This template provided a mechanical stop for the outward movements and is referred to as the 'stopper ring'. Direct vision of hand and stylus was prevented by a horizontal opaque board. The position of the stylus was recorded and mapped online to the position of a (visible or invisible) cursor on the monitor using Matlab with the Psychophysics Toolbox extension [36].

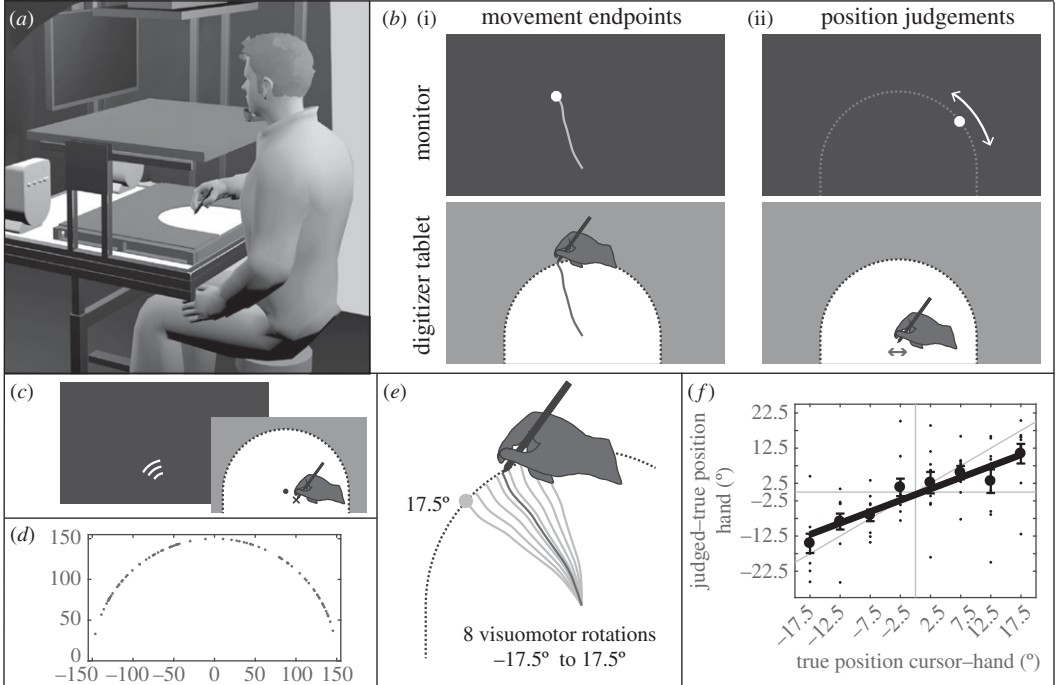

**Figure 2.** Experimental set-up and task. (*a*) Participants moved their hand within a half-circular workspace (white area) a horizontal digitizer tablet. The corresponding cursor movements could be shown on a frontoparallel monitor. The hand was always occluded. (*b*) Movements were made from the centre of the workspace to its boundary and back. Trajectories of the hand and cursor (dark and light grey lines, respectively) are shown here for illustration purposes only. Participants were to remember the most outward position of the hand and cursor, referred to as the movement endpoint. After returning to the centre position, participants moved a cursor along the invisible track of possible endpoints (here indicated by the white dotted line) by making small left-right hand movements to report the position judgements. (*c*) Before movement onset, we displayed a WiFi-like symbol to instruct participant which approximate direction their outward movements should be directed to. The hand was in an initial position, somewhat below the centre position, when the symbol was shown. (*d*) Due to the instructed approximate movement direction, endpoints of the hand movements were spread out rather than stereotypical. (*e*) The movements of the cursor were rotated relative to the direction of the hand movement by one of eight possible visuomotor rotations. The thus caused discrepancies in the hand and cursor movement endpoints were needed to assess the biases in the hand and cursor position judgements. (*f*) We regressed the judgement error (i.e. the deviation between the judged and true endpoints) on the true angular difference between the hand and cursor endpoints (i.e. the visuomotor rotation), here indicated for hand position judgements. The slope of this regression indicates the bias of the hand position judgements away from the true position (the horizontal line) and towards the 'irrelevant' cursor position (the diagonal line). The variance over the residuals indicates the variability of the position judgements.

## 2.3. Task and procedure

Participants performed out-and-back movements with their right hand on the digitizer tablet. Regarding the outward movement, they were instructed to start at the centre of the semicircular workspace and move comfortably fast in an indicated outward direction until hitting the stopper ring. Regarding the backward movement, they were instructed to return back to the remembered start position and to do so immediately upon hitting the stopper ring. They confirmed the end of the backward movement by pressing the button on the stylus. A cursor, a filled white circle of 6 mm diameter, provided visual feedback of the movement on the computer monitor. The cursor was visible at the movement endpoint (i.e. during the short period at which the hand was static at the stopper ring). In addition, it was visible either during the outward or the backward hand movement, depending on the experimental condition. The hand was occluded during all phases of the movement and in all conditions.

After each out-and-back movement, the word 'HAND' or 'CURSOR' appeared on the monitor to instruct participants to report the final position of, respectively, the hand or the cursor at the end of the outward movement, that is, where it hit the stopper ring (figure 2*b*(i)). After 500 ms, a visual marker (6 mm diameter white dot) appeared at the far left or far right side of an invisible semicircular track covering all possible endpoints of the outward movements (i.e. stopper ring in monitor space). Participants controlled the velocity of the marker along the track by making small movements to the

left or right (less than 1 cm amplitude) with the stylus on the tablet (figure 2b(ii)). Once satisfied with the marker's position, they confirmed their judgement by pressing the stylus's button. There were no time constraints in making these position judgements.

In order to prevent stereotyped directions of the outward movements and hence stereotyped position judgements, each trial was started as follows: we first guided participants to an initial position by means of an arrow pointing to that position. The initial position was randomly chosen in a rectangular area ranging between 1 to 2 cm below and between −2 to 2 cm to the side of the centre position. When the initial position was reached, a visual indicator similar to a WiFi symbol was presented on the monitor for 1 s (figure 2c). It served to instruct participants to move approximately in one of eight movement directions in each trial (−56° to 56° relative to straight ahead, in steps of 16°). As a result of the instructed approximate movement directions, the movement endpoints were scattered over the entire stopper ring (see figure 2d for the endpoints of an illustrative participant). Next, both the centre position (a 7 mm diameter open circle) and the cursor were shown. As participants reached the centre position, the open circle and—depending on the visibility condition—the cursor disappeared and a beep indicated the start of the outward movement. Specific details on, for example, the size and relative timing of all displayed items can be found in Debats et al. ([8], Experiment 2).

## 2.4. Visuomotor rotations

The direction of cursor motion was rotated relative to the direction of hand movement in each trial. We thus created a small discrepancy between the physical hand and cursor endpoints. This visuomotor rotation varied randomly across trials between −17.5° and +17.5° in steps of 5°, with a mean of 0° (figure 2e). Participants were not informed about its presence (they were told that the experiment was about the effect of attention on remembered sensations), and none of the participants reported noticing it in a structured post-experimental interview. For cursor-control tasks, it is fairly typical that small and random visuomotor discrepancies, and sometimes even considerable discrepancies, remain unnoted (e.g. [10,37–39]). The small discrepancies between hand and cursor endpoints allowed us to assess the biases in the reported hand positions towards the physical cursor positions and the biases in the reported cursor positions towards the physical hand positions.

## 2.5. Visibility conditions

We compared two experimental conditions. In condition *Out*, the cursor was shown during the outward movement and during the short period in which the hand was stationary in the end position, but not during the backward movement. In condition *Back*, the cursor was shown at the endpoint and during the backward movement, but not during the outward movement. Thus, in both conditions, the cursor was shown in the final position that participants later on had to judge, but in the condition *Out* it was shown in addition before the final position was reached, whereas in the condition *Back* it was shown in addition after the final position had been reached. Thus, correlations between the kinematics of hand and cursor movements could be experienced either before or after reaching the endpoint, the position of which had to be judged, but in both conditions before the judgements were actually provided.

## 2.6. Trial types

We tested two bimodal types of trials, which served to assess the strength of sensory integration, and two unimodal types from which we estimated parameters that were needed for model predictions. In the bimodal trials, participants made the out-and-back hand movements while watching the corresponding cursor movement on the monitor (when the cursor was shown depended on the visibility condition). After participants had returned their hand to the remembered centre position, they judged either the final hand position (*BiHand* trials) or the final cursor position (*BiCursor* trials) of the outward movement. In one unimodal trial type, participants made the out-and-back hand movements without the cursor being shown, after which they judged the final hand position (*UniHand* trials). In the other unimodal trial type, participants kept their hand static at the centre position while watching a cursor during either an outward or a backward movement (depending on the visibility condition) as well as in the end position (*UniCursor* trials). In these trials, the cursor motion was a replay of a cursor trajectory recorded in a preceding *BiCursor* trial. Participants were instructed to judge the final cursor position after a delay that was equal to the average duration of the return movements in the preceding trials.

## 2.7. Design

The experimental trials differed with respect to two visibility conditions, four trial types and eight visuomotor rotations. In the unimodal trials (*UniHand* and *UniCursor* trials), the visuomotor rotation was a dummy variable, that is, it was coded for these trials, but in *UniHand* trials the cursor was invisible and in *UniCursor* trials there was no hand movement; the replayed cursor trajectory was from a *BiCursor* trial with the corresponding visuomotor rotation. The two visibility conditions were assigned to separate experimental sessions, and their order was counterbalanced across participants. For each visibility condition, the set of 32 trials (four trial types × eight visuomotor rotations) was repeated 10 times, resulting in a total of 320 trials per session. Per repetition set, the order of the 32 trials was semi-randomized with the constraint that each *UniCursor* trial occurred later in the sequence than the corresponding *BiCursor* trial (because the cursor trajectory presented in the *UniCursor* trials was recorded in the corresponding *BiCursor* trial). For the trials of each repetition set, we randomly allocated one of the eight visuomotor rotations to one of the eight instructed approximate movement directions, thus preventing a systematic relation between movement direction and rotation. Each experimental session was preceded by up to 28 familiarization trials, during which the experimenter gave verbal instructions on the required task execution. The familiarization ended when participants were able to correctly execute the task. We organized the total number of 320 trials per session (without familiarization) into six blocks with short breaks in between. Each session was performed on a separate day and took about 2.5 h to complete.

## 2.8. Data preprocessing

The movement endpoints were defined as the positions in which the tip of the stylus in the hand and/or the centre of the cursor on the monitor had covered 97% of the 150 mm radial movement distance between the centre position and the stopper ring. This 3% margin was required because, due to the thickness of the stylus and depending on the orientation of the stylus in the hand, the maximum radial distance varied over trials, ranging between 97% and 100%. For each trial, we recorded the Cartesian coordinates of the physical cursor and/or hand movement endpoints as well as the coordinates of the judged hand or cursor position. These Cartesian coordinates were converted into angles of a polar coordinate system with the origin in the centre position, that is, in the start position of the outward movements.

In a first step, the data were screened for outliers. A trial was classified as an outlier when (i) the direction of outward movement deviated more than 35° from the instructed direction, (ii) the absolute angular deviation between the physical and judged position was larger than 35°, which is twice the maximal visuomotor rotation, and (iii) the hand or cursor moved more than 2.5° to the left or right after reaching the stopper ring (or the equivalent distance on the monitor). For each participant, there were 80 trials per trial type and condition (8 rotations × 10 repetitions). One participant had up to 48 of these 80 trials identified as outliers. This participant, who apparently did not follow the task instructions, was excluded from all analyses. For the other participants, between 0 and 11 trials were identified as outliers.

In a second step, the judgements were corrected for hysteresis effects. Motions of the visual marker, the position of which was matched to the remembered position of the cursor or hand at the end of the outward movement, started either at the far left or right of the semicircular track. Systematically different judgements with these different initial positions could inflate the observed variances of the judgements. To remove such differences, for each experimental session (i.e. 320 trials), we computed the regression of the judged direction on the physical direction with the constraint of a single slope, but two different intercepts for marker motions starting on the left or right side. The judged directions were corrected for the difference between the intercepts (which ranged from −4.2° to +2.7°) by adding or subtracting half the intercept difference.

## 2.9. Dependent variables—observed

Following these initial steps of data analysis, we assessed the integration strength as well as the biases and variability of the judgements from the bimodal trials (*BiHand* and *BiCursor* trials). We also estimated the variability of the unisensory hand and cursor position judgements from the unimodal trials in order to allow the generation of model predictions (see below). Additionally, we computed

three kinematic measures: the duration of the outward movement, the time at the endpoint and the duration of the backward movement.

The proportional biases of the bimodal position judgements were determined from the regressions of the judgement error observed in each trial (i.e. the angular deviation of the judged cursor or hand endpoint from the respective physical endpoint) on the visuomotor rotation (i.e. the angular deviation of the physical cursor endpoint from that of the hand). These regressions were computed separately for each participant, each experimental condition, and for the *BiCursor* and *BiHand* trials. An example is illustrated in figure 2f. The slopes of these regressions quantify the degree to which judgements of the cursor endpoint were biased to the hand endpoint, and vice versa. More specifically, the slopes quantify the perceptual biases as proportions of the visuomotor rotation. In a weighted average, they correspond to the weight given to the other modality [8]. For the *BiHand* trials, it is the weight $w_{C\_obs}$ of the cursor's end position, and for the *BiCursor* trials it is the weight $w_{H\_obs}$ of the hand's end position

$$J_{H.C} = J_H + w_{C\_obs}(J_C - J_H) = (1 - w_{C\_obs})J_H + w_{C\_obs}J_C,$$
$$J_{C.H} = J_C + w_{H\_obs}(J_H - J_C) = (1 - w_{H\_obs})J_C + w_{H\_obs}J_H,$$

where $J_{H.C}$ and $J_{C.H}$ are the mean judgements of hand and cursor positions in *BiHand* and *BiCursor* trials, respectively, whereas $J_H$ and $J_C$ are the mean judgements of hand and cursor positions in *UniHand* and *UniCursor* trials. Replacing the mean judgements by the sums of the stimulus positions and the mean judgement errors $\Delta$, we get: $J_{H.C} = s_H + \Delta_{H.C}$, $J_{C.H} = s_C + \Delta_{C.H}$, $J_H = s_H + \Delta_H$ and $J_C = s_C + \Delta_C$. Rearranging terms results in the regression equations used to compute the weights $w_{H\_obs}$ and $w_{C\_obs}$

$$\Delta_{H.C} = \Delta_H + w_{C.obs}(\Delta_C - \Delta_H) + w_{C\_obs}(s_C - s_H),$$
$$\Delta_{C.H} = \Delta_C + w_{H.obs}(\Delta_H - \Delta_C) - w_{H\_obs}(s_C - s_H),$$

where $(s_C - s_H)$ is the visuomotor rotation. When the weights $w_{H\_obs} = w_{C\_obs} = 0$, judgements of the cursor's and hand's endpoints in bimodal trials match the corresponding judgements in unimodal trials, but when the weights $w_{H\_obs} = w_{C\_obs} = 1$, judgements of the cursor's and hand's end position in bimodal trials match the hand's and cursor's position, respectively, in unimodal trials.

The integration strength was defined as the sum of the bias weights, that is: $\lambda_{obs} = w_{H\_obs} + w_{C\_obs}$. An integration strength of 1 indicates full sensory integration, that is, judgements of cursor and hand positions in bimodal trials become identical, $J_{H.C} = J_{C.H}$. An integration strength of 0 indicates independence (i.e. hand and cursor positions are not biased towards each other). Any strength between 0 and 1 indicates partial integration (i.e. a certain level of mutual biases in the hand and cursor position judgements). It should be noted that numerically the integration strength was, in our analyses, not constrained to range between 0 and 1 because we determined $w_{H\_obs}$ and $w_{C\_obs}$ independently.

The variances of the bimodal position judgements were estimated as the variances of the residuals of the regressions for the *BiHand* trials ($\sigma_{H.\_obs}$) and for the *BiCursor* trials ($\sigma_{C.H\_obs}$). To estimate the unisensory variances, we performed a similar regression analysis as for the bimodal trials. We regressed the judgement errors of the unimodal position estimates on the dummy variable visuomotor rotation and likewise computed the variances of the residuals for the *UniCursor* trials ($\sigma_{C\_obs}$) and for the *UniHand* trials ($\sigma_{H\_obs}$).

Finally, we computed the durations of the outward and backward movements, as well as the time spent at the endpoint. The endpoint duration was defined as the time between the outward hand movement passing 97% of the 150 mm radial distance between the centre position and the stopper ring, and the backward movement falling below this 97% radial distance. The duration of the outward hand movement was defined as the time between 3 and 97% of this 150 mm radial distance. The duration of the backward movement was defined as the time between the 97% threshold and a threshold of 4.5 mm (i.e. the equivalent of 3% of 150 mm) from the position in which participants reported being back in the centre position.

## 2.10. Dependent variables—model predictions

The experiment was designed to reveal potential differences in integration strength as a consequence of the different time windows of causality information in the two experimental conditions. To reveal potential additional effects of the manipulation, we generated model predictions for partial integration when based on a purely reliability-based weighting mechanism. We used an extension of the classical model for optimal integration, which allows for partial integration (cf. [8–10,29,35]). This so-called coupling-prior model, proposed by Ernst [13,22,40], describes partial integration as a weighted sum of

the unisensory estimates whereby the weights can add up to any value between 0 (no integration) and 1 (full integration, or fusion) due to an additional parameter called the variance of the coupling prior. This variance of the coupling prior represents the strength of the causality judgement, the belief that two sensory signals belong together (see e.g. [20]). It can only be visualized in a particular two-dimensional illustration of integration. But, as such an illustration requires lengthy explanation, we refer to our previous work in which this is provided [8]. Note that the term 'prior' might lead to confusion here; the coupling prior could be confused with a sensory prior (i.e. previously collected sensory experience that has shaped an expectation—a prior—regarding the content of the redundant sensory signals (e.g. [41,42]), or it could be confused with a causality prior (i.e. previously collected evidence regarding the causal relation between two sensory signals). Instead, the coupling prior, and its variance in particular, reflects the combined causality judgement that is based on all available causality evidence. We considered the model suitable for the present purpose because it can generate predictions for the bimodal position judgements' characteristics (i.e. proportional biases and variability) based on the input of integration strength and the two unisensory variabilities. As in the 'classical' full integration model, this model is based on the principle of reliability-based weighting.

The model predictions were derived as follows (cf. [8]). First, for each condition and each participant, we computed the theoretical variance of the coupling prior $\sigma^2_{\text{prior}}$ that reflects the observed integration strength ($\lambda_{\text{obs}}$)

$$\sigma^2_{\text{prior}} = \frac{1 - \lambda_{\text{obs}}}{\lambda_{\text{obs}}} \; (\sigma^2_{\text{C\_obs}} + \; \sigma^2_{\text{H\_obs}}).$$

Using this parameter in the following equations, we computed—for the observed integration strength—the optimal-integration predictions for the magnitudes of the biases in the bimodal position judgements:

$$w_{\text{H\_pred}} = \frac{\sigma^2_{\text{C\_obs}}}{\sigma^2_{\text{C\_obs}} + \; \sigma^2_{\text{H\_obs}} + \; \sigma^2_{\text{prior}}}$$

and

$$w_{\text{C\_pred}} = \frac{\sigma^2_{\text{H\_obs}}}{\sigma^2_{\text{C\_obs}} + \; \sigma^2_{\text{H\_obs}} + \; \sigma^2_{\text{prior}}},$$

and the optimal-integration predictions for the variances of the bimodal position judgements

$$\sigma^2_{\text{C.H\_pred}} = \frac{\sigma^2_{\text{C\_obs}}(\sigma^2_{\text{H\_obs}} + \; \sigma^2_{\text{prior}})}{\sigma^2_{\text{C\_obs}} + \; \sigma^2_{\text{H\_obs}} + \; \sigma^2_{\text{prior}}}$$

and

$$\sigma^2_{\text{H.C\_pred}} = \frac{\sigma^2_{\text{H\_obs}}(\sigma^2_{\text{C\_obs}} + \; \sigma^2_{\text{prior}})}{\sigma^2_{\text{C\_obs}} + \; \sigma^2_{\text{H\_obs}} + \; \sigma^2_{\text{prior}}}.$$

## 2.11. Statistical analysis

We first tested our dependent measures for deviations from normality using Shapiro–Wilk tests. When violations were observed, we used non-parametric tests (Wilcoxon signed-rank tests); otherwise, we used parametric tests (*t*-tests, ANOVAs and Pearson correlations). For the variability of the position judgements, we used standard deviations instead of variances because their unit (degrees of arc rather than squared degrees of arc) can be more easily interpreted. Values reported in the text and figures are means ± standard errors except when stated otherwise.

# 3. Results

One participant was excluded from the analyses because he/she had too many outlier trials to reliably compute the dependent variables (see the Methods section). One other participant was identified as an outlier in that he/she violated the task instruction to return to the start position directly upon hitting the workspace boundary. This participant stayed at the endpoint for 1557 and 910 ms in conditions *Out* and *Back*, respectively. For the remaining 12 participants, these mean durations at the endpoint were 496 ± 42

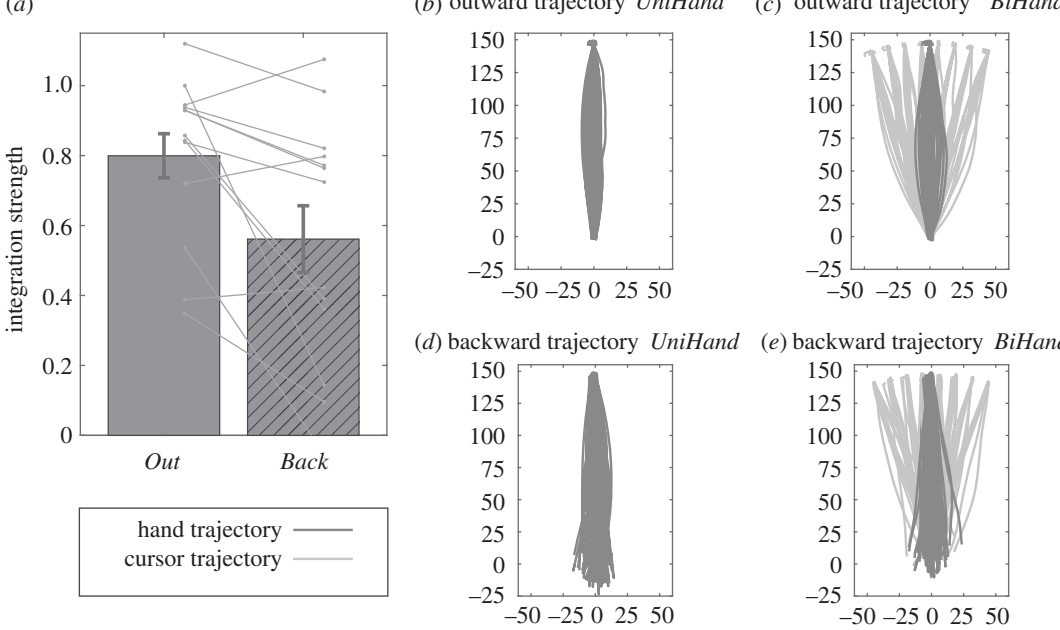

**Figure 3.** Integration strength and movement trajectories. (*a*) The group mean integration strength for the *Out* and *Back* visibility conditions. The light grey lines indicate individual participants. (*b*) The outward hand movement trajectories for an exemplary participant in the condition *Out* when no cursor was presented (i.e. *UniHand* trials). The actual trajectory endpoints were scattered over the workspace boundary, but in order to illustrate the variability in the trajectories, we here rotated all trajectories to align the hand endpoints at 90°. (*c*) The outward trajectories of the hand movement (dark grey lines) and the corresponding cursor movement (light grey lines) illustrated here for the *BiHand* trials of the same exemplary participant. Again, the trajectories were rotated to align the hand endpoints at 90°. The corresponding cursor trajectories now clearly illustrate the eight levels of the visuomotor rotation. (*d,e*) Same as (*b,c*), but now for the backward movement in the condition *Back*.

and $535 \pm 66$ ms. The outlier participant thus had a longer period to obtain the redundant hand and cursor position information. As this participant did not appear as an outlier in the dependent variables regarding the sensory integration, and did not affect the statistical significance of the tests performed, he/she was included in the final sample.

## 3.1. Integration strength

The mean strength of the sensory integration in the two experimental conditions is shown in figure 3*a*. This is the sum of the proportional bias (or bias weight) in the hand position judgements towards the true cursor position and the proportional bias (or bias weight) in the judged cursor position towards the true hand position. Sensory integration was stronger when the cursor was visible during the outward movement (condition *Out*) than when it was visible during the backward movement (condition *Back*). This difference was statistically significant, as indicated by a paired-samples *t*-test ($t_{12} = 3.00$, $p = 0.011$).

Although the strength of sensory integration was reduced in condition *Back*, some level of integration persisted. This remaining integration strength could be based on the causality evidence provided during the backward movement, and/or on other sources of causality information. These sources could be the causality prior, and possibly additional causality information being present when the hand and cursor are both in the endpoint, at different endpoints across trials. To assess this, we compared the current data with those of a previous study in which the cursor was visible only at the to-be-judged endpoint, so neither during the forward movement nor during the backward movement [29]. In the baseline condition of that study, which was identical to the current condition *Out*, the mean integration strength was $0.68 \pm 0.10$. When the cursor was visible in the endpoint only, the integration strength was $0.47 \pm 0.11$. This compares to the current findings where the integration strength reduced from $0.80 \pm 0.07$ in the condition *Out* to $0.58 \pm 0.10$ in the condition *Back* with the causality information present during the backward movement. We conducted a two (experiment) × two (condition: baseline versus manipulation) ANOVA with the experiment as a between-participant factor and condition as a

within-participant factor. Whereas the main effect of condition was significant ($F_{1,22} = 13.67$, $p = 0.001$), both the main effect of experiment ($F_{1,22} = 0.785$, $p = 0.385$) and the interaction ($F_{1,22} = 0.01$, $p = 0.940$) were not significant. According to this analysis, the remaining integration strength was equivalent in the condition with causality information during the backward movement and a condition with no causality information from the correlated hand–cursor trajectories at all.

## 3.2. Movement characteristics

The causality evidence that supports the hand–cursor integration was present in the correlated hand–cursor movement trajectories. Hence, we compared the outward and backward trajectories and their durations as markers for the quality and quantity of the causality evidence in the *Out* and *Back* experimental conditions.

Figure 3b–e illustrates the movement trajectories of a representative participant. For the condition *Out*, we show the outward paths (figure 3b,c) and for the condition *Back* the backward paths (figure 3d,e). The dark grey lines indicate the hand trajectories, and light grey lines the cursor trajectories. Figure 3b,d shows the *UniHand* trials, and thus hand trajectories only (dark grey lines). Figure 3c,e shows the *BiHand* trials and thus both hand and cursor trajectories (light grey lines). The trajectories of the hand and cursor started in the common central position and ended scattered over the full range of the stopper ring (figure 2d). The trajectories shown in figure 3 were rotated to align the hand endpoints at 90°. The corresponding cursor trajectories now clearly reveal the eight different visuomotor rotations (figure 3c,e). Note that the movement endpoint was defined at 97% of the radius of the semicircular workspace, in order to account for the small variation across trials in the maximal radial movement distance. In figure 3, the small blobs at the workspace boundary illustrate the small excess of the trajectories beyond this 97% threshold.

Visual inspection of the outward trajectories in the condition *Out* (figure 3b,c) and the backward trajectories in the condition *Back* (figure 3d,e) reveals a clear difference in that the outward trajectories start at the centre position, whereas the backward trajectories did not end at the actual centre position, but at the position where participants remembered the start position to be. Importantly, however, the cross-correlations between hand and cursor trajectories are neither affected by the exact paths of the movements, nor by backward movements not ending at the centre position. The quality of the causality evidence provided by cross-correlations was thus equal for the two visibility conditions.

The duration of the outward and backward trajectories indicates the time during which the correlated hand and cursor movements—and thus causality evidence—were available in the bimodal trials. This is a marker of the quantity of the causality evidence. In the condition *Out*, the duration of the outward movements, as averaged over participants and *BiCursor* and *BiHand* trials, was $1417 \pm 229$ ms, and in the condition *Back*, the mean duration of backward movements was $1318 \pm 130$ ms. The difference was not significant according to a paired-samples *t*-test ($t_{12} = 0.65$, $p = 0.526$). Additionally, there was no inter-individual correlation between the movement durations and the integration strength as indicated by the correlations (*Out*: $r = 0.19$, $p = 0.535$; *Back*: $r = 0.01$, $p = 0.978$).

## 3.3. Characteristics of the position judgements

The reduced integration strength in the condition *Back* indicates that the causality evidence was less effective in this condition. To examine whether the integration was in other ways affected by the experimental manipulation, we analysed the characteristics of the unimodal and the biased bimodal position judgements and compared them against model predictions as a benchmark for optimal partial integration. The specific model we used predicts for predefined integration strength, the magnitude of the biases and the variability of the bimodal judgements that would result from purely reliability-based weighting of the unisensory information. It does not model the causal inference stage; it only predicts the bimodal characteristics based on the unisensory variabilities and desired integration strength (with the integration strength reflecting the outcome of the causal inference stage).

First, we show the unimodal variabilities (figure 4a) as quantified by the standard deviation of the position judgements in the unimodal trials. These variabilities are the input to the model predictions. Variability was clearly smaller for the judgements of the cursor position (*UniCursor* trials; light grey bars) than for the judgements of the hand position (*UniHand* trials; dark grey bars). Furthermore, the variability was highly similar between the visibility conditions *Out* (solid bars) and *Back* (striped bars), as confirmed by non-significant differences (Wilcoxon signed-rank tests) both for *UniCursor* trials ($T_{13} = 48$, $z = 0.18$, $p = 0.861$) and *UniHand* trials ($T_{13} = 30$, $z = 1.08$, $p = 0.279$). For the *UniHand*

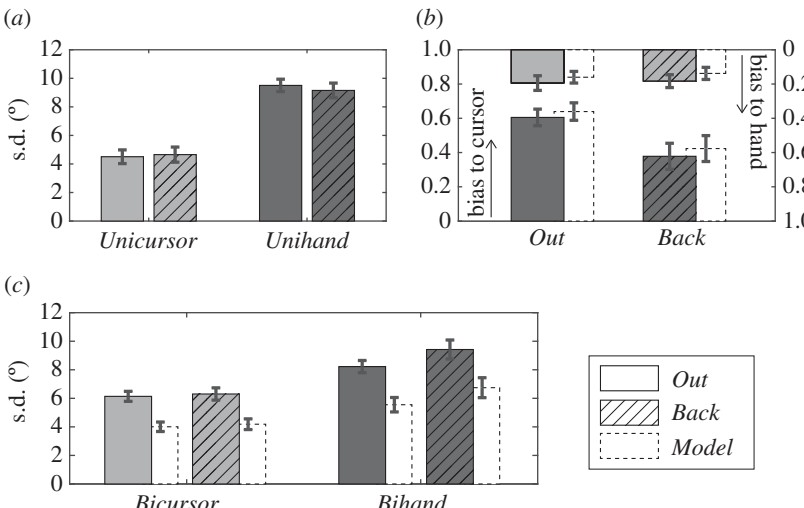

**Figure 4.** Observed and predicted position judgements' characteristics. Condition *Out* is indicated by the solid grey bars, condition *Back* by the striped grey bars and model prediction by the open white bars. Dark grey colours indicate the judgements of the hand position; light grey colours the judgements of the cursor position. (*a*) The group variability—quantified by the standard deviation—of the unimodal position judgements. (*b*) The biases in the hand position judgements towards the position of the cursor (standing bars) or the biases in the cursor position judgements towards the position of the hand (hanging bars). (*c*) The variability of the bimodal position judgements as quantified by the standard deviation.

judgements, this result was as expected, given that these trials did not differ between the two visibility conditions. For the *UniCursor* judgements, this result indicates that the variability of the judgements of cursor endpoints was not affected by the cursor being visible during the trajectory towards this position versus away from it.

Figure 4*b* shows the biases of the bimodal position judgements as upward bars (*BiHand* trials; bias of judged hand position towards the cursor position) and downward bars (*BiCursor* trials; bias of judged cursor position towards the hand position). Note that these biases are the proportional biases as determined over all levels of rotation (see the Methods section and figure 2*f*), not the absolute judgement errors that differed for the various levels of visuomotor rotation. As the integration strength was defined as the sum of these upward and downward bars, the difference between the visibility conditions in integration strength is now visible by the different 'gaps' between the bars. The figure shows that the reduced integration strength in the condition *Back* was mostly due to a substantially reduced proportional bias in the judged hand position towards the cursor position ($T_{13} = 15$, $z = 2.13$, $p = 0.033$; Wilcoxon signed-rank test). The difference between conditions *Out* and *Back* in the proportional bias of the judged cursor position towards the hand position was only small and not statistically significant ($T_{13} = 41$, $z = 0.31$, $p = 0.753$). This pattern of results was consistent with the model predictions for partial integration (the dotted open bars), in that the reduction of integration strength in the condition *Back* should be most strongly reflected in the position judgements with the highest unimodal variability.

Finally, figure 4*c* shows the observed variability (filled grey bars) of the biased bimodal position judgements. For the condition *Back*, we found a significantly increased variability of the hand position judgements ($T_{13} = 78$, $z = 2.27$, $p = 0.023$; Wilcoxon signed-rank test), but not of the cursor position judgements ($T_{13} = 55$, $z = 0.66$, $p = 0.507$; Wilcoxon signed-rank test). This effect was consistent with the model predictions (dotted open bars) in that the model predicts that the lower integration strength in conditions *Back* should lead to stronger increased variability for the position judgements with the highest unimodal variability. The figure also shows that the predicted variability was substantially lower than the observed variability, an observation that confirms previous findings, as will be discussed below.

# 4. Discussion

In the current study, we compared the effectiveness of causality evidence that precedes the to-be-judged sensory information with that of causality evidence that directly follows the to-be-judged sensory

information. The to-be-judged sensory information in our task is the endpoint of a reaching movement. The causality evidence that links the hand and cursor in this task is available in the hand–cursor-correlated movement trajectories. We could selectively provide this information before the to-be-judged reach endpoint by showing the cursor only during the outward hand movement (condition *Out*). Similarly, we could provide the causality information selectively after the to-be-judged reach endpoint by showing the cursor only during the backward hand movement (condition *Back*). We quantified the strength of the integration, which directly reflects the effectiveness of the underlying causality evidence. Our data clearly revealed stronger integration in the condition *Out* than in the condition *Back*, indicating that causality evidence available before obtaining the to-be-judged movement endpoint information has a stronger effect than causality evidence obtained thereafter.

Although the strength of sensory integration was reduced in the condition *Back*, some level of integration persisted. Is this remaining strength of integration based on the causality evidence provided during the backward movement? The tentative answer to this question is 'no' because in previous studies we have observed about the same strength of integration when the cursor was only visible at the endpoint, that is, at the position to be judged, but neither during the forward movement nor during the backward movement [29]. Thus, there is currently no evidence to suggest that the remaining integration strength in the condition *Back* was due to causality evidence obtained during the backward movement.

The two experimental conditions of the present experiment had been designed to provide the same simultaneous hand position and cursor position information at the endpoint of the outward movements, but to provide causality evidence either before or after the endpoints are reached. However, there is a different perspective on the difference between the two conditions. Although angular differences between cursor and hand positions in the polar coordinate system centred in the start position of the outward movements were constant during each outward or backward movement, the linear distances increased and decreased, respectively. This could have affected the biases observed, provided that the judgements were not only based on the perceived positions of cursor and hand at the endpoint, but during a larger part of the outward and backward movements. In addition, if the perceived positions of hand and cursor were influenced by an average of the experienced linear distances during the outward or backward movements, the additional assumption of a stronger weight of earlier linear distances than that of later ones in the average would be required. Then, the judged positions of cursor and hand at the end of the outward movements could be closer together than the judged positions of cursor and hand at the start of the backward movements—which could result in observing stronger and weaker biases, respectively. If this perspective on the present findings were correct, the reduction of the biases and thus of coupling strength in the condition *Back* as compared with the condition *Out* of the present experiment should have been stronger than in the previous study [29] when the cursor was presented at the endpoint only. This was clearly not the case. In spite of no support for this alternative perspective on our findings, a definite rejection might require a dedicated experiment with constant linear rather than angular distances between the positions of hand and cursor during each movement.

What does our finding, that effective causality evidence seems to be collected until presentation of the to-be-judged sensory information, but hardly or not at all thereafter, imply for our understanding of the subsequent integration process? Our findings are in line with the notion that the to-be-judged sensory information is integrated as soon as it is available and no longer once it is no longer available (here, when leaving the movement endpoint). With this scenario, causality evidence that is presented after the to-be-judged sensory information must be ineffective because the unisensory estimates can no longer be accessed. This notion is at odds with a hypothesis of Rand & Heuer [4], proposing that sensory information on hand and cursor positions is integrated only once the judgement is required. That hypothesis was based on the finding that, in a similar experimental paradigm as the one of the present study, a pause of 6 s at the endpoint with no visual feedback caused an increased bias of judged cursor position towards the position of the hand and a reduced bias of judged hand position towards the position of the cursor. This change in the relative magnitude of the biases is consistent with an increase in the relative reliability of the hand position information over the additional time period during which hand position information is obtained. However, these previous findings do not conflict with the conclusion that integration no longer takes place once the hand has left the to-be-judged position. They merely demonstrate that integration does not end when visual cursor position information on the endpoint is no longer available, but hand position information still is. Jointly these findings suggest that both causal inference and sensory integration are completed as soon as no more

sensory information on the to-be-judged attribute is available. In the current experimental paradigm and the quite similar one of Rand & Heuer [4], this was as soon as the backward movement had started.

The main focus of this study was on the strength of the integration, that is, on the sum of the proportional biases in the bimodal position judgements. In order to reveal potential additional effects of the experimental manipulation, we analysed further characteristics of the bimodal position judgements. This revealed that in the condition *Back*, the bimodal judgements of the hand position showed a strongly reduced proportional bias as well as an increased variability, whereas the judgements of the cursor position showed these effects to a much smaller and statistically not significant degree. Does this mean that our manipulation only influenced the integration of hand and cursor position information in the case that judgements of the hand position are requested? Or did the late causality evidence in the condition *Back* lead to a reduced attention to hand position information? Neither seems to be the case. Rather, a basic model of partial optimal integration (cf. [13,22,40,43]) predicts exactly this pattern of results as a consequence of the higher variability of unimodal hand position judgements than of unimodal cursor position judgements. The observed change in integration strength thus was reflected predominantly in reduced biases in the bimodal hand position judgements (with corresponding changes in judgement variability). Yet this effect is not bound to the proprioceptive modality *per se*, but results from it being the more variable sensory information in this task. The model predictions thus strongly suggest that the experimental manipulation, making causality information available either before or after the to-be-judged sensory information, had no other effects on the integration of hand and cursor positions than those that followed from the reduced effectiveness of the causality evidence.

In the current study, as well as in our previous studies [8,9,29], we have compared the characteristics of the biased position judgements with the predictions of a model of optimal partial integration proposed by Ernst (cf. [13,22,40,43]). This so-called coupling-prior model provides predictions for purely reliability-based weighting, as the classic model for optimal integration (i.e. sensory fusion, or maximum-likelihood estimation) does, yet with integration strengths smaller than one. The model was consistently found to accurately account for the magnitude of the hand–cursor mutual biases. Yet it was also consistently found to predict a systematically lower variability of the biased position judgements. The origin of the consistent discrepancy between observed and predicted standard deviations is not resolved, but it seems likely that there is a source of variability that is not adequately considered by the model or that is not adequately captured by the variability in unimodal trials. The purpose of including model comparisons in the current study was not to find a best-fitting model for the observed behaviour. Instead, we aimed to provide a benchmark for interpreting the combined effect of different integration strengths given different unisensory reliabilities.

In sensorimotor control, it is important for the brain to identify the visual signals that result from one's own actions. Multisensory causal inference seems to be a neural mechanism through which this can be achieved. Causal inference could thus serve sensorimotor control by underlying the enhanced temporal processing of visual information regarding one's own actions referred to as visuomotor binding [1], the temporal attraction between an action and its visual or auditory consequence referred to as intentional binding (e.g. [44,45]), as well as the sense of agency and body ownership (e.g. [46,47]). However, the identification of causal relations between actions and their visual effect suffices for these purposes. There is no purpose for a subsequent multisensory integration in tasks such as cursor control, given that such integration comes at the cost of biases and a decreased ability to perceive sensory discrepancies. The consistent observation of such biases in the typical cursor-control task studied here thus may suggest that sensory integration is an obligatory consequence of causal inference rather than an optional one.

In summary, the current study reveals that causality judgements that link our actions (here, a hand movement) to their spatially distant visual consequences (here, a cursor motion) are based on causality evidence (here, correlated hand–cursor movements) that is obtained before the to-be-judged sensory information is presented (here, a movement endpoint), whereas that same causality evidence seems to be essentially ineffective when obtained after the to-be-judged sensory information.

Ethics. The experiment was conducted in accordance with the declaration of Helsinki and approved by the Bielefeld University ethics committee (the 'Ethik-Kommission der Universität Bielefeld') with approval number EUB 2015-059. All participants gave written informed consent prior to their participation.

Data accessibility. All data generated and analysed in this study are available at pub.uni-bielefeld.de/data/2930664, with doi:10.4119/unibi/2930664.

Authors' contributions. N.B.D. and H.H. designed and conducted the experiment, analysed the data and wrote the manuscript.

Competing interests. The authors declare having no competing interests.

Funding. The contribution of N.B.D. was supported by the German Research Foundation (DFG) grant no. HE1187/19-1. We furthermore acknowledge the financial support of the German Research Foundation and the Open Access Publication Fund of Bielefeld University for the article processing charges.

Acknowledgements. We thank A. Oppenborn for assistance in the data collection.

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
