## [Reviewer comments · Royal Society Open Science]

Review History

RSOS-181429.R0 (Original submission)

Review form: Reviewer 1

Is the manuscript scientifically sound in its present form?

No

Are the interpretations and conclusions justified by the results?

No

Is the language acceptable?

Yes

Is it clear how to access all supporting data?

Yes

Do you have any ethical concerns with this paper?

No

Have you any concerns about statistical analyses in this paper?

No

Recommendation?

Major revision is needed (please make suggestions in comments)

Comments to the Author(s)

In this paper the authors address the timely and important question of causal inference underlying multisensory integration. It is a prominent theory in the field that redundant sensory cues are combined in a statistically optimal manner, but the basic underlying assumption, that all the cues have the same cause or convey information about the same variable, has been overlooked. Here it is suggested that the nervous system performs causal inference prior to combining or not sensory information. The topic is interesting, however I admit that the paper as is did not convince me. The methodological section is difficult to follow at times, making it difficult to follow how the claims are supported by the data. More importantly there is a conspicuous lack of documentation on behaviour, such that the authors jump from illustrative traces (Fig. 2) to summary variables (Fig. 3) leaving too little intuition for the reader to understand the authors' reasoning. Likewise the model is too succinctly described to understand the results presented in Figure 3.

Aside of these general issues there is one fundamental step that appears problematic. It is correct that an optimal cue combination is achieved with weighting factors that sum to 1 (λ obs, page 14) when there is no prior (maximum likelihood combination). But here there is a prior, and it appears in the equations page 15. It is visible on page 15 that the weights do not sum to 1 due to the prior. I apologise if I missed something but in my view the problem is that departure from optimality, captured by the integration strength or by the standard deviations, are taken as a proxy of a supposed causal inference process, while it could relate to the unknown statistics of the internal prior. Before reworking data presentation and clarifying the argument it is critical to demonstrate that causal inference can be dissociated from another phenomenon, and I admit I do not think that it is true. That is, assume that there is no inference in the nervous system and that participants always considered that vision and limb feedback came from the same source, isn't it possible to account for the results by changing the prior only? I realize it is an ad hoc interpretation, but the problem is that we know the nervous system uses priors, and learns to change these priors under changing conditions.

Specific comments:

The abstract is unclear and difficult to follow in the first pass. We do not know anything about the paradigm at this stage thus wording like "to-be-judged information", and mention of out and back movements induce confusion.

Page 3 line ~33: "In the current study,...". Pls clarify, how did the analyses focus on a specific time window?

Page 4, 3rd paragraph: Causality is an assumption and cannot be established with correlational data. Too often the manuscript hints on that. If signals are correlated then an observer, or the brain, may assume that there is a causal link.

In general I found the introduction was quite long and sounded a little unfocused.

Page 10, second paragraph: Based on this paragraph it appears that the cursor was shown during movement and in the end for both out and back movements. What was the difference?

Page 12, second paragraph: It is unclear why the participant with many outliers was excluded. What was his/her behaviour? If he/she could not aim in the prescribed window of angles, it may be due to motor noise leaving unchanged the process of multisensory integration. If the subject has behaviour consistent with the other I do not see why he/she should be excluded. It is also

questionable why another participant was excluded based on timing requirement. It seems that the time spent for this participant was about or less than three times the duration for the other. Does that really impacted his/her behaviour?

Page 13: Pls give quantitative information about the removing of the bias.

Figure 2: Pls describe more accurately the spatial alignment procedure. Were the start and end points of each trace aligned to vertical? Is that why the traces all end at the same location? As is the figure creates the impression of a very high accuracy constraint. Why was it not the case for backwards trajectories?

Figure 3: Somehow the visuomotor rotations disappeared from the pictures and the behaviour was summarised in bar graphs. Pls provide a more detailed analysis of data across rotations, subjects, etc.

Review form: Reviewer 2

Is the manuscript scientifically sound in its present form?

Yes

Are the interpretations and conclusions justified by the results?

No

Is the language acceptable?

Yes

Is it clear how to access all supporting data?

No

Do you have any ethical concerns with this paper?

No

Have you any concerns about statistical analyses in this paper?

No

Recommendation?

Major revision is needed (please make suggestions in comments)

Comments to the Author(s)

This study investigates the temporal window of integration of actions and their visual consequences. Based on the results of a behavioral task and computational modeling, the authors conclude that only information presented before the to-be-judged movement endpoint affect the causal inference process.

This is an interesting paper; the experiment is properly performed, and the data analyses is convincing. That being said, there are a few points that need to be addressed in order to fully support the main conclusion.

1. The spatial offset between the hand and the cursor was introduces as a rotation, so that the more participants moved away from the starting point, the larger was the conflict between hand and cursor position. As a consequence, outward trials started with no offset and ended with maximum offset, whereas the backward trials started with maximum offset and ended with not offset. Could this difference (conflict at the beginning vs. the end) alone explain the difference in

the strength of integration between the two conditions? A potential control for this would be to invert the effect of the rotation, so that is maximal at the start position, and absent at the endpoint.

2. The fact that weights that do not sum to one necessarily implies incomplete integration should be properly explained. Also, in principle one could expect to have other sources of information (e.g., a prior or acoustic information) each of them weighted based on their reliability and fused together according to MLE: in this case the weights of the cursor and the hand would not sum to one but there could still be complete fusion, right? It would be nice to state clearly why the sum of the weights univocally (at least according to the chosen model) represents the strength of integration. Moreover, the model should be fully described, including how the weights determine the bias (i.e., weighted sum).

3. This paper fully relies on a specific model for weak fusion, however there are several alternative models. One in particular, the Causal Inference model (Körding et al. 2007), would be particularly relevant as it explicitly estimates the probability of different sources of sensory information having a common cause (plus causal inference is also mentioned in the title of the current paper). It would be useful to see how this alternative model fares with the data and run a model comparison to identify which of them better explains human behavior.

4. The reliability of hand and cursor position is measured from the judgment errors; however, such a metric does not purely reflect perceptual noise, but it also includes motor noise. Such an additional source of noise may be the factor underlying the discrepancy between data and model predictions in Figure 3C (the authors also discuss about additional sources of errors in the discussion). It would be nice to see how the model would fare once such an additional source of noise is taken into account (e.g., leaving the variance of motor noise as a free parameter; assuming motor noise to be additive to perceptual noise such a source could then be subtracted from the judgment error variance).

Minor points:

P. 4: For completeness, the term “correspondence problem” could be added here.

P. 5: end of first paragraph. This sentence is not clear: how is the strength of integration measured, and how is that different from measuring the strength of causality judgment?

P. 5: last parag: The description of this previous study is a bit cryptic

P. 20: “only causality evidence available before obtaining the to-be-judged movement endpoint information has an effect [...]”. This sounds like a rather strong claim given that there was integration also in the back condition (as clearly stated by the authors themselves in the next paragraph of the discussion).

P. 22: “integration does not end before the maximum amount of sensory information is obtained”. Please support this claim with evidence (e.g., how is the maximum amount of information measured?)

Decision letter (RSOS-181429.R0)

11-Dec-2018

Dear Dr Debats,

The editors assigned to your paper ("Optimal integration of actions and their visual effects: exploring the time window for multisensory causal inference") have now received comments from reviewers. We would like you to revise your paper in accordance with the referee and Associate Editor suggestions which can be found below (not including confidential reports to the Editor). Please note this decision does not guarantee eventual acceptance.

Please submit a copy of your revised paper before 03-Jan-2019. Please note that the revision deadline will expire at 00.00am on this date. If we do not hear from you within this time then it will be assumed that the paper has been withdrawn. In exceptional circumstances, extensions may be possible if agreed with the Editorial Office in advance. We do not allow multiple rounds of revision so we urge you to make every effort to fully address all of the comments at this stage. If deemed necessary by the Editors, your manuscript will be sent back to one or more of the original reviewers for assessment. If the original reviewers are not available, we may invite new reviewers.

- Data accessibility

If you wish to submit your supporting data or code to Dryad (<http://datadryad.org/>), or modify your current submission to dryad, please use the following link:
<http://datadryad.org/submit?journalID=RSOS&manu=RSOS-181429>

- Competing interests

- Authors' contributions

- Acknowledgements

- Funding statement

on behalf of Dr Atsushi Iriki (Associate Editor) and Antonia Hamilton (Subject Editor)
openscience@royalsociety.org

Comments to Author:

Reviewers' Comments to Author:

Reviewer: 1

Comments to the Author(s)

In this paper the authors address the timely and important question of causal inference underlying multisensory integration. It is a prominent theory in the field that redundant sensory cues are combined in a statistically optimal manner, but the basic underlying assumption, that all the cues have the same cause or convey information about the same variable, has been overlooked. Here it is suggested that the nervous system performs causal inference prior to combining or not sensory information. The topic is interesting, however I admit that the paper as is did not convince me. The methodological section is difficult to follow at times, making it difficult to follow how the claims are supported by the data. More importantly there is a conspicuous lack of documentation on behaviour, such that the authors jump from illustrative traces (Fig. 2) to summary variables (Fig. 3) leaving too little intuition for the reader to

understand the authors' reasoning. Likewise the model is too succinctly described to understand the results presented in Figure 3.

Aside of these general issues there is one fundamental step that appears problematic. It is correct that an optimal cue combination is achieved with weighting factors that sum to 1 (lambda obs, page 14) when there is no prior (maximum likelihood combination). But here there is a prior, and it appears in the equations page 15. It is visible on page 15 that the weights do not sum to 1 due to the prior. I apologise if I missed something but in my view the problem is that departure from optimality, captured by the integration strength or by the standard deviations, are taken as a proxy of a supposed causal inference process, while it could relate to the unknown statistics of the internal prior. Before reworking data presentation and clarifying the argument it is critical to demonstrate that causal inference can be dissociated from another phenomenon, and I admit I do not think that it is true. That is, assume that there is no inference in the nervous system and that participants always considered that vision and limb feedback came from the same source, isn't it possible to account for the results by changing the prior only? I realize it is an ad hoc interpretation, but the problem is that we know the nervous system uses priors, and learns to change these priors under changing conditions.

Specific comments:

The abstract is unclear and difficult to follow in the first pass. We do not know anything about the paradigm at this stage thus wording like "to-be-judged information", and mention of out and back movements induce confusion.

Page 3 line ~33: "In the current study,...". Pls clarify, how did the analyses focus on a specific time window?

Page 4, 3rd paragraph: Causality is an assumption and cannot be established with correlational data. Too often the manuscript hints on that. If signals are correlated then an observer, or the brain, may assume that there is a causal link.

In general I found the introduction was quite long and sounded a little unfocused.

Page 10, second paragraph: Based on this paragraph it appears that the cursor was shown during movement and in the end for both out and back movements. What was the difference?

Page 12, second paragraph: It is unclear why the participant with many outliers was excluded. What was his/her behaviour? If he/she could not aim in the prescribed window of angles, it may be due to motor noise leaving unchanged the process of multisensory integration. If the subject has behaviour consistent with the other I do not see why he/she should be excluded. It is also questionable why another participant was excluded based on timing requirement. It seems that the time spent for this participant was about or less than three times the duration for the other. Does that really impacted his/her behaviour?

Page 13: Pls give quantitative information about the removing of the bias.

Figure 2: Pls describe more accurately the spatial alignment procedure. Were the start and end points of each trace aligned to vertical? Is that why the traces all end at the same location? As is the figure creates the impression of a very high accuracy constraint. Why was it not the case for backwards trajectories?

Figure 3: Somehow the visuomotor rotations disappeared from the pictures and the behaviour was summarised in bar graphs. Pls provide a more detailed analysis of data across rotations, subjects, etc.

Reviewer: 2

Comments to the Author(s)

This study investigates the temporal window of integration of actions and their visual consequences. Based on the results of a behavioral task and computational modeling, the authors conclude that only information presented before the to-be-judged movement endpoint affect the causal inference process.

This is an interesting paper; the experiment is properly performed, and the data analyses is convincing. That being said, there are a few points that need to be addressed in order to fully support the main conclusion.

1. The spatial offset between the hand and the cursor was introduced as a rotation, so that the more participants moved away from the starting point, the larger was the conflict between hand and cursor position. As a consequence, outward trials started with no offset and ended with maximum offset, whereas the backward trials started with maximum offset and ended with not offset. Could this difference (conflict at the beginning vs. the end) alone explain the difference in the strength of integration between the two conditions? A potential control for this would be to invert the effect of the rotation, so that is maximal at the start position, and absent at the endpoint.

2. The fact that weights that do not sum to one necessarily implies incomplete integration should be properly explained. Also, in principle one could expect to have other sources of information (e.g., a prior or acoustic information) each of them weighted based on their reliability and fused together according to MLE: in this case the weights of the cursor and the hand would not sum to one but there could still be complete fusion, right? It would be nice to state clearly why the sum of the weights univocally (at least according to the chosen model) represents the strength of integration. Moreover, the model should be fully described, including how the weights determine the bias (i.e., weighted sum).

3. This paper fully relies on a specific model for weak fusion, however there are several alternative models. One in particular, the Causal Inference model (Körding et al. 2007), would be particularly relevant as it explicitly estimates the probability of different sources of sensory information having a common cause (plus causal inference is also mentioned in the title of the current paper). It would be useful to see how this alternative model fares with the data and run a model comparison to identify which of them better explains human behavior.

4. The reliability of hand and cursor position is measured from the judgment errors; however, such a metric does not purely reflect perceptual noise, but it also includes motor noise. Such an additional source of noise may be the factor underlying the discrepancy between data and model predictions in Figure 3C (the authors also discuss about additional sources of errors in the discussion). It would be nice to see how the model would fare once such an additional source of noise is taken into account (e.g., leaving the variance of motor noise as a free parameter; assuming motor noise to be additive to perceptual noise such a source could then be subtracted from the judgment error variance).

Minor points:

P. 4: For completeness, the term “correspondence problem” could be added here.

P. 5: end of first paragraph. This sentence is not clear: how is the strength of integration measured, and how is that different from measuring the strength of causality judgment?

P. 5: last parag: The description of this previous study is a bit cryptic

P. 20: “only causality evidence available before obtaining the to-be-judged movement endpoint

information has an effect [...]". This sounds like a rather strong claim given that there was integration also in the back condition (as clearly stated by the authors themselves in the next paragraph of the discussion).

P. 22: "integration does not end before the maximum amount of sensory information is obtained". Please support this claim with evidence (e.g., how is the maximum amount of information measured?)

Author's Response to Decision Letter for (RSOS-181429.R0)

See Appendix A.

RSOS-181429.R1 (Revision)

Review form: Reviewer 1

Is the manuscript scientifically sound in its present form?

No

Are the interpretations and conclusions justified by the results?

No

Is the language acceptable?

Yes

Is it clear how to access all supporting data?

Yes

Do you have any ethical concerns with this paper?

No

Have you any concerns about statistical analyses in this paper?

No

Recommendation?

Reject

Comments to the Author(s)

The authors have made extensive changes to the manuscript and took most concerns previously raised into account. However, I admit I still have substantial reservations about the paper. The main issue in my view is that a change in integration strength, admitting it is measurable based on behaviour, is not unequivocal proof of changes in causality inference because the weights also depend on the prior which is unknown. I expressed this concern in my earlier review and I do not feel that the authors have demonstrated that their data could say anything about causality inference. I trust the authors have carefully considered my earlier concern and I am willing to reconsider if there is a flaw in my own reasoning, but as such I cannot endorse the study.

The experiments show that multisensory integration is different if the information is provided before or after the event that they are asked to report. This result is sound and interesting, but any claim about causality inference overstretches the data. Indeed, it is possible that the results

simply reflect a time-dependent integration process, and that when a judgment is made, sensory information provided after has less impact because estimation is not performed backwards. Thus, the prior is given more importance if sensory information is given later. Causality inference in my view is not necessary to explain this data. In fact, the authors appear to mix the likelihood and a putative inference of causality. That is if the likelihood of feedback is high dependent on their own internal model they will certainly rely strongly on it because the data is correlated with the prediction, but this is not causality inference. If my own reasoning is flawed then perhaps insight from a third angle is needed to arbitrate.

Overall the description of the experiment has much improved. There remain problems with the theoretical part: I failed to understand the way the model was used as it seems to me that equations linking λ and the sigmas on page 18 comes from standard maximum likelihood model that does consider a common source for each piece of information. What I failed to understand then was how this was used to assess causality inference since causality was previously assumed to derive these equations. In addition, the way predictions were derived was unclear: it is written that the predicted gap was set equal to the observed gap, but in this case the authors performed a post hoc fitting of the data and these are not real model predictions. There is no mechanism to explain how integration strength should evolve prior to seeing the data. I was left with the impression that the model had no explanatory power.

Specific points:

First three lines of page 6 are almost impossible to follow.

Middle of page 6: the question is also unnecessarily complicated. Why not just recast the paper in terms of asking whether the temporal ordering between sensory information and perception matters?

The theoretical framework is missing internal feedback which may or may not be included as a prior, but which is definitely used in the brain.

Anovas in the discussion is strange, this could be moved to Results.

Review form: Reviewer 2

Is the manuscript scientifically sound in its present form?

Yes

Are the interpretations and conclusions justified by the results?

Yes

Is the language acceptable?

Yes

Is it clear how to access all supporting data?

Yes

Do you have any ethical concerns with this paper?

No

Have you any concerns about statistical analyses in this paper?

Yes

Recommendation?

Accept with minor revision (please list in comments)

Comments to the Author(s)

Overall, the authors have done a good job at addressing my comments. I only have a few remaining comments:

1. Looking at Figure 4C there seems to be a clear difference between observed and predicted variability, however this does not reach significance as assessed by the ANOVA.
2. "Figure 3B" in the second paragraph of page 22 should be "Figure 4B"
3. Most of the dataset analyzed here are unlikely to be normal (i.e., standard deviations can only be positive, relative biases and integration strength take value-approximately-between 0 and 1, etc..) however they are analyzed using parametric tests that assume an underlying normal distribution. It would be nice to test for normality and if systematic deviations occur run appropriate transformation or choose different tests (though I guess this might not dramatically change the conclusions).

Decision letter (RSOS-181429.R1)

30-May-2019

Dear Dr Debats:

Manuscript ID RSOS-181429.R1 entitled "Exploring the time window for causal inference and the multisensory integration of actions and their visual effects" which you submitted to Royal Society Open Science, has been reviewed. The comments from reviewer(s) are included at the bottom of this letter.

In view of the criticisms of the reviewer(s), I must decline the manuscript for publication in Royal Society Open Science at this time. However, a new manuscript may be submitted which takes into consideration these comments.

Please note that resubmitting your manuscript does not guarantee eventual acceptance, and that your resubmission will be subject to re-review by the reviewer(s) before a decision is rendered.

You will be unable to make your revisions on the originally submitted version of your manuscript. Instead, revise your manuscript using a word processing program and save it on your computer.

You may also click the below link to start the resubmission process (or continue the process if you have already started your resubmission) for your manuscript. If you use the below link you will not be required to login to ScholarOne Manuscripts.

*** PLEASE NOTE: This is a two-step process. After clicking on the link, you will be directed to a webpage to confirm. ***

https://mc.manuscriptcentral.com/rsos?URL_MASK=4577afdaf0264397975e089973c473b6

Because we are trying to facilitate timely publication of manuscripts submitted to Royal Society Open Science, your resubmitted manuscript should be submitted by 27-Nov-2019. If you are unable to submit by this date please contact the Editorial Office for options.

I look forward to a resubmission.

Kind regards,
 Alice Power
 Royal Society Open Science
 openscience@royalsociety.org

on behalf of Dr Atsushi Iriki (Associate Editor) and Antonia Hamilton (Subject Editor)
 openscience@royalsociety.org

Reviewer comments to Author:
 Reviewer: 1

Comments to the Author(s)

The authors have made extensive changes to the manuscript and took most concerns previously raised into account. However, I admit I still have substantial reservations about the paper. The main issue in my view is that a change in integration strength, admitting it is measurable based on behaviour, is not unequivocal proof of changes in causality inference because the weights also depend on the prior which is unknown. I expressed this concern in my earlier review and I do not feel that the authors have demonstrated that their data could say anything about causality inference. I trust the authors have carefully considered my earlier concern and I am willing to reconsider if there is a flaw in my own reasoning, but as such I cannot endorse the study.

The experiments show that multisensory integration is different if the information is provided before or after the event that they are asked to report. This result is sound and interesting, but any claim about causality inference overstretches the data. Indeed, it is possible that the results simply reflect a time-dependent integration process, and that when a judgment is made, sensory information provided after has less impact because estimation is not performed backwards. Thus, the prior is given more importance if sensory information is given later. Causality inference in my view is not necessary to explain this data. In fact, the authors appear to mix the likelihood and a putative inference of causality. That is if the likelihood of feedback is high dependent on their own internal model they will certainly rely strongly on it because the data is correlated with the prediction, but this is not causality inference. If my own reasoning is flawed then perhaps insight from a third angle is needed to arbitrate.

Overall the description of the experiment has much improved. There remain problems with the theoretical part: I failed to understand the way the model was used as it seems to me that equations linking λ and the σ s on page 18 comes from standard maximum likelihood model that does consider a common source for each piece of information. What I failed to understand then was how this was used to assess causality inference since causality was previously assumed to derive these equations. In addition, the way predictions were derived was unclear: it is written that the predicted gap was set equal to the observed gap, but in this case the authors performed a post hoc fitting of the data and these are not real model predictions. There is no mechanism to explain how integration strength should evolve prior to seeing the data. I was left with the impression that the model had no explanatory power.

Specific points:

First three lines of page 6 are almost impossible to follow.

Middle of page 6: the question is also unnecessarily complicated. Why not just recast the paper in terms of asking whether the temporal ordering between sensory information and perception matters?

The theoretical framework is missing internal feedback which may or may not be included as a prior, but which is definitely used in the brain.

Anovas in the discussion is strange, this could be moved to Results.

Reviewer: 2

Comments to the Author(s)

Overall, the authors have done a good job at addressing my comments. I only have a few remaining comments:

1. Looking at Figure 4C there seems to be a clear difference between observed and predicted variability, however this does not reach significance as assessed by the ANOVA.
2. "Figure 3B" in the second paragraph of page 22 should be "Figure 4B"
3. Most of the dataset analyzed here are unlikely to be normal (i.e., standard deviations can only be positive, relative biases and integration strength take value-approximately-between 0 and 1, etc..) however they are analyzed using parametric tests that assume an underlying normal distribution. It would be nice to test for normality and if systematic deviations occur run appropriate transformation or choose different tests (though I guess this might not dramatically change the conclusions).

Author's Response to Decision Letter for (RSOS-181429.R1)

See Appendix B.

RSOS-192056.R0

Review form: Reviewer 2

Is the manuscript scientifically sound in its present form?

Yes

Are the interpretations and conclusions justified by the results?

Yes

Is the language acceptable?

Yes

Do you have any ethical concerns with this paper?

No

Have you any concerns about statistical analyses in this paper?

No

Recommendation?

Accept as is

Comments to the Author(s)

I am satisfied with the current version of the manuscript

Review form: Reviewer 3

Is the manuscript scientifically sound in its present form?

Yes

Are the interpretations and conclusions justified by the results?

No

Is the language acceptable?

Yes

Do you have any ethical concerns with this paper?

No

Have you any concerns about statistical analyses in this paper?

No

Recommendation?

Major revision is needed (please make suggestions in comments)

Comments to the Author(s)

See attachment (Appendix C).

Decision letter (RSOS-192056.R0)

20-Feb-2020

Dear Dr Debats,

The Subject Editor assigned to your paper ("Exploring the time window for causal inference and the multisensory integration of actions and their visual effects") has now received comments from reviewers. We would like you to revise your paper in accordance with the referee and Associate Editor suggestions which can be found below (not including confidential reports to the Editor). Please note this decision does not guarantee eventual acceptance.

Please submit a copy of your revised paper before 14-Mar-2020. Please note that the revision deadline will expire at 00.00am on this date. If we do not hear from you within this time then it will be assumed that the paper has been withdrawn. In exceptional circumstances, extensions may be possible if agreed with the Editorial Office in advance. We do not allow multiple rounds of revision so we urge you to make every effort to fully address all of the comments at this stage. If deemed necessary by the Editors, your manuscript will be sent back to one or more of the original reviewers for assessment. If the original reviewers are not available we may invite new reviewers.

When submitting your revised manuscript, you must respond to the comments made by the referees and upload a file "Response to Referees" in "Section 6 - File Upload". Please use this to document how you have responded to each of the comments, and the adjustments you have made. In order to expedite the processing of the revised manuscript, please be as specific as possible in your response.

- Ethics statement

- Data accessibility

<http://datadryad.org/submit?journalID=RSOS&manu=RSOS-192056>

- Competing interests

- Authors' contributions

- Acknowledgements

- Funding statement

Kind regards,
Andrew Dunn

Royal Society Open Science Editorial Office
Royal Society Open Science
openscience@royalsociety.org

on behalf of Dr Atsushi Iriki (Associate Editor) and Antonia Hamilton (Subject Editor)
openscience@royalsociety.org

Reviewer comments to Author:
Reviewer: 2

Comments to the Author(s)
I am satisfied with the current version of the manuscript

Reviewer: 3

Comments to the Author(s)
See attachment

Author's Response to Decision Letter for (RSOS-192056.R0)

See Appendix D.

RSOS-192056.R1 (Revision)

Review form: Reviewer 2

Is the manuscript scientifically sound in its present form?
Yes

Are the interpretations and conclusions justified by the results?
Yes

Is the language acceptable?
Yes

Do you have any ethical concerns with this paper?
No

Have you any concerns about statistical analyses in this paper?
No

Recommendation?
Accept as is

Comments to the Author(s)
The paper reads well and there are no standing concerns

Review form: Reviewer 3

Is the manuscript scientifically sound in its present form?

Yes

Are the interpretations and conclusions justified by the results?

No

Is the language acceptable?

Yes

Do you have any ethical concerns with this paper?

No

Have you any concerns about statistical analyses in this paper?

No

Recommendation?

Accept with minor revision (please list in comments)

Comments to the Author(s)

See attachment (Appendix E).

Decision letter (RSOS-192056.R1)

Dear Dr Debats:

On behalf of the Editors, I am pleased to inform you that your Manuscript RSOS-192056.R1 entitled "Exploring the time window for causal inference and the multisensory integration of actions and their visual effects" has been accepted for publication in Royal Society Open Science subject to minor revision in accordance with the referee suggestions. Please find the referees' comments at the end of this email.

The reviewers and Subject Editor have recommended publication, but also suggest some minor revisions to your manuscript. Therefore, I invite you to respond to the comments and revise your manuscript.

- **Ethics statement**

- **Data accessibility**

It is a condition of publication that all supporting data are made available either as supplementary information or preferably in a suitable permanent repository. The data accessibility section should state where the article's supporting data can be accessed. This section should also include details, where possible of where to access other relevant research materials

such as statistical tools, protocols, software etc can be accessed. If the data has been deposited in an external repository this section should list the database, accession number and link to the DOI for all data from the article that has been made publicly available. Data sets that have been deposited in an external repository and have a DOI should also be appropriately cited in the manuscript and included in the reference list.

<http://datadryad.org/submit?journalID=RSOS&manu=RSOS-192056.R1>

- **Competing interests**

- **Authors' contributions**

- **Acknowledgements**

- **Funding statement**

Because the schedule for publication is very tight, it is a condition of publication that you submit the revised version of your manuscript before 03-Jul-2020. Please note that the revision deadline will expire at 00.00am on this date. If you do not think you will be able to meet this date please let me know immediately.

When submitting your revised manuscript, you will be able to respond to the comments made by the referees and upload a file "Response to Referees" in "Section 6 - File Upload". You can use this

to document any changes you make to the original manuscript. In order to expedite the processing of the revised manuscript, please be as specific as possible in your response to the referees.

Kind regards,

Anita Kristiansen
Editorial Coordinator

on behalf of Dr Atsushi Iriki (Associate Editor)
openscience@royalsociety.org

Reviewer comments to Author:
Reviewer: 3

Comments to the Author(s)
See attachment ("review_2.pdf")

Reviewer: 2

Comments to the Author(s)
The paper reads well and there are no standing concerns

Author's Response to Decision Letter for (RSOS-192056.R1)

See Appendix F.

Decision letter (RSOS-192056.R2)

Dear Dr Debats,

It is a pleasure to accept your manuscript entitled "Exploring the time window for causal inference and the multisensory integration of actions and their visual effects" in its current form for publication in Royal Society Open Science. The comments of the reviewer(s) who reviewed your manuscript are included at the foot of this letter.

on behalf of Dr Atsushi Iriki (Associate Editor)
openscience@royalsociety.org

Appendix A

Comments to Author:

Reviewer: 1

Comments to the Author(s)

In this paper the authors address the timely and important question of causal inference underlying multisensory integration. It is a prominent theory in the field that redundant sensory cues are combined in a statistically optimal manner, but the basic underlying assumption, that all the cues have the same cause or convey information about the same variable, has been overlooked. Here it is suggested that the nervous system performs causal inference prior to combining or not sensory information. The topic is interesting, however I admit that the paper as is did not convince me. The methodological section is difficult to follow at times, making it difficult to follow how the claims are supported by the data. More importantly there is a conspicuous lack of documentation on behaviour, such that the authors jump from illustrative traces (Fig. 2) to summary variables (Fig. 3) leaving too little intuition for the reader to understand the authors' reasoning. Likewise the model is too succinctly described to understand the results presented in Figure 3.

We have revised parts of the methods and results section to improve readability and clarity. We now provided a more detailed descriptions of the data analyses, the dependent variables, and we provided a more elaborate explanation of the model used. Furthermore, we have added a figure (now Figure 2), to illustrate the model.

Aside of these general issues there is one fundamental step that appears problematic. It is correct that an optimal cue combination is achieved with weighting factors that sum to 1 (λ obs, page 14) when there is no prior (maximum likelihood combination). But here there is a prior, and it appears in the equations page 15. It is visible on page 15 that the weights do not sum to 1 due to the prior. I apologise if I missed something but in my view the problem is that departure from optimality, captured by the integration strength or by the standard deviations, are taken as a proxy of a supposed causal inference process, while it could relate to the unknown statistics of the internal prior. Before reworking data presentation and clarifying the argument it is critical to demonstrate that causal inference can be dissociated from another phenomenon, and I admit I do not think that it is true. That is, assume that there is no inference in the nervous system and that participants always considered that vision and limb feedback came from the same source, isn't it possible to account for the results by changing the prior only? I realize it is an ad hoc interpretation, but the problem is that we know the nervous system uses priors, and learns to change these priors under changing conditions.

We are sorry for not having been more explicit on the nature (or function) of the coupling prior in the model; this is changed now (p. 16-17). The variance of the coupling prior is that ingredient of the model that captures the strength of integration.

We also state explicitly now (p. 4) that nearly all models that relate causal evidence to sensory coupling posit a monotonic relation between the two.

Specific comments:

The abstract is unclear and difficult to follow in the first pass. We do not know anything about the paradigm at this stage thus wording like "to-be-judged information", and mention of out and back movements induce confusion.

We have rewritten the abstract to enhance clarity.

Page 3 line ~33: "In the current study,...". Pls clarify, how did the analyses focus on a specific time window?

We added an explanatory sentence which specifies the time windows compared.

Page 4, 3rd paragraph: Causality is an assumption and cannot be established with correlational data. Too often the manuscript hints on that. If signals are correlated then an observer, or the brain, may assume that there is a causal link.

That is definitely correct. However, it is equally correct that correlations are taken as evidence of a common cause (or a common source) of stimulation although, by a strict logic, a common cause does imply correlated signals, but correlated signals do not imply a common cause/source. In fact, in the cursor-control task integration might even result from a "misinterpretation" of kinematic correlations: when one sees and feels movements of one's hand, kinematic correlations correctly indicate a common cause/source of the signals, but when one feels movements of one's hand and sees motion of a cursor, the same perfect correlations do not indicate a common source. This would be equivalent to all perceptual mechanisms that work fine in most situations, but lead to illusions in other situations (mostly unusual ones that one normally does not encounter or that appeared only late during evolution, as the control of cursor positions). We agree that this is an interesting issue, but it is clearly beyond the purpose of this study, so we thought it better not to add a discussion of this point.

In general I found the introduction was quite long and sounded a little unfocused.

We have revised parts of the introduction to make the line of thought clearer.

Page 10, second paragraph: Based on this paragraph it appears that the cursor was shown during movement and in the end for both out and back movements. What was the difference?

We have expanded the description of the two conditions to make the difference crystal clear (p. 10, second paragraph): the cursor was shown in the end and during the outward movement in one condition and in the end and during the backward movement in the other condition.

Page 12, second paragraph: It is unclear why the participant with many outliers was excluded. What was his/her behaviour? If he/she could not aim in the prescribed window of angles, it may be due to motor noise leaving unchanged the process of multisensory integration. If the subject has behaviour consistent with the other I do not see why he/she should be excluded. It is also questionable why another participant was excluded based on timing requirement. It seems that the time spent for this participant was about or less than three times the duration for the other. Does that really impacted his/her behaviour?

We consider our criteria for detecting outlier trials as being rather mild. In the current experimental task, motor noise cannot account for errors of over 35 degrees in the movement direction, or in the position judged. It is more likely that in such trials, the participant did not pay attention to the instructions shown on the monitor. After exclusion of these outlier trials, one participant had too little remaining trials to reliably compute the dependent variables and thus had to be excluded. The other

participant was excluded because he/she did not follow the instruction to return immediately upon hitting the workspace boundary. We have reconsidered this exclusion criterion and now included this participant in the final sample. Doing so did not affect the significance of the statistical tests performed. This rationale is provided at the beginning of the results section (p. 18).

Page 13: Pls give quantitative information about the removing of the bias.

The range of the differences between the intercepts are given now (p. 13).

Figure 2: Pls describe more accurately the spatial alignment procedure. Were the start and end points of each trace aligned to vertical? Is that why the traces all end at the same location? As is the figure creates the impression of a very high accuracy constraint. Why was it not the case for backwards trajectories?

The actual trajectories were scattered over the full range of the half-circular bow, and indeed, they were aligned here for clarity purposes. We have added a few sentences in the main text in which we explain the spatial alignment procedure that was used to generate these graphs (p. 19).

Figure 3: Somehow the visuomotor rotations disappeared from the pictures and the behaviour was summarised in bar graphs. Pls provide a more detailed analysis of data across rotations, subjects, etc.

The biases presented in Figure 3 are not the absolute judgement errors (these indeed differ for the various levels of visuomotor rotation), but the relative bias as determine over the data for all levels of visuomotor rotation. The detail of how the bias is computed are now described more elaborately in the methods section, and in the results section we added a statement about why the visuomotor rotation is not a factor in these biases (p. 20).

Reviewer: 2

Comments to the Author(s)

This study investigates the temporal window of integration of actions and their visual consequences. Based on the results of a behavioral task and computational modeling, the authors conclude that only information presented before the to-be-judged movement endpoint affect the causal inference process.

This is an interesting paper; the experiment is properly performed, and the data analyses is convincing. That being said, there are a few points that need to be addressed in order to fully support the main conclusion.

1. The spatial offset between the hand and the cursor was introduces as a rotation, so that the more participants moved away from the starting point, the larger was the conflict between hand and cursor position. As a consequence, outward trials started with no offset and ended with maximum offset, whereas the backward trials started with maximum offset and ended with not offset. Could this difference (conflict at the beginning vs. the end) alone explain the difference in the strength of integration between the two conditions? A potential control for this would be to invert the effect of the rotation, so that is maximal at the start position, and absent at the endpoint.

This is an interesting perspective on the two conditions of the experiment. We have added a paragraph in the discussion on it. Of course, this concern only holds when the offset is not measured as an angle in the polar coordinate system, but as a distance in a Cartesian coordinate system. And it presupposes that the judgments are based on a temporal average, with stronger weight for earlier than for later spatial offsets. A stringent control would require a constant linear offset during each movement (there must be an offset at the position that is judged, otherwise biases cannot be determined, and this cannot be the start position – both forward and backward movements either follow or precede it). However, even without such a control condition, as described in the manuscript, the reduction of coupling strength with the cursor presented during the backward movement should be stronger than with the cursor presented at the endpoint only, which is not the case according to our data (admittedly from two different experiments).

2. The fact that weights that do not sum to one necessarily implies incomplete integration should be properly explained. Also, in principle one could expect to have other sources of information (e.g., a prior or acoustic information) each of them weighted based on their reliability and fused together according to MLE: in this case the weights of the cursor and the hand would not sum to one but there could still be complete fusion, right? It would be nice to state clearly why the sum of the weights univocally (at least according to the chosen model) represents the strength of integration. Moreover, the model should be fully described, including how the weights determine the bias (i.e., weighted sum).

We have added the equations on p. 14 which indicate how the weights measure the influence of the position of cursor or hand that currently is not the judged position. These equations also show that the weights can be estimated from the regressions (and are proportional biases), and that full fusion is reached when they add up to 1, so that in bimodal trials cursor and hand are judged to be in the same position. Note that these equations are based only on the simple assumption of a weighted average of the positions of hand and cursor, so there is not yet a model behind these equations (except one would designate the assumption of a weighted average as a model).

3. This paper fully relies on a specific model for weak fusion, however there are several alternative models. One in particular, the Causal Inference model (Körding et al. 2007), would be particularly relevant as it explicitly estimates the probability of different sources of sensory information having a common cause (plus causal inference is also mentioned in the title of the current paper). It would be useful to see how this alternative model fares with the data and run a model comparison to identify which of them better explains human behavior.

The primary purpose of this study is not a comparison of models. There are now several versions of the BCI model proposed by Körding et al (model averaging, model selection, probability matching), that including such a comparison would require additional explanation regarding the differences between them. This would shift the focus of the manuscript away from what we intend it to be. We chose to use the coupling prior model for the simple reason that we have done so in previous studies, as stated now on p. 16. This model is simple in that it has only one parameter that needs to be set (i.e., that the predicted integration strength matches the observed strength), and it doesn't require optimization procedures that depend on initial settings.

4. The reliability of hand and cursor position is measured from the judgment errors; however, such a metric does not purely reflect perceptual noise, but it also includes motor noise. Such an additional source of noise may be the factor underlying the discrepancy between data and model predictions in Figure 3C (the authors also discuss about additional sources of errors in the discussion). It would be nice to see how the model would fare once such an additional source of noise is taken into account (e.g., leaving the variance of motor noise as a free parameter; assuming motor

noise to be additive to perceptual noise such a source could then be subtracted from the judgment error variance).

This is likely one of the first ideas that come to mind when faced with the discrepancy between observed and predicted variances. We have tried such an account, as described in Debats et al., JNP, 2017, but including such a term in fact increases the difference between predicted and observed variance. The discrepancy between predicted and observed variances is a robust phenomenon that we have observed in all our previous studies. We have examined whether it is related to the two reference frames used (horizontal for hand, vertical for the cursor and for the judgments). This however, was found not to affect the variances. We suspect that an additional source of noise is involved, and are planning dedicated research to address this issue in future work.

Minor points:

P. 4: For completeness, the term "correspondence problem" could be added here.

Done (p. 3).

P. 5: end of first paragraph. This sentence is not clear: how is the strength of integration measured, and how is that different from measuring the strength of causality judgment?

We have introduced the principle of measuring the strength of integration as "sum of mutual biases", details are given later in the manuscript. The outcome of the causality judgment is mainly a theoretical variable, which, as we state now, can be assessed via integration strength as a proxy (p. 4 first paragraph). We believe that the revised introduction should be clear in this respect.

P. 5: last parag: The description of this previous study is a bit cryptic

This part has been rewritten.

P. 20: "only causality evidence available before obtaining the to-be-judged movement endpoint information has an effect [...]". This sounds like a rather strong claim given that there was integration also in the back condition (as clearly stated by the authors themselves in the next paragraph of the discussion).

We have tempered this statement (p. 23).

P. 22: "integration does not end before the maximum amount of sensory information is obtained". Please support this claim with evidence (e.g., how is the maximum amount of information measured?)

Sentence is changed to make its meaning clearer (p. 26)

Appendix B

Reviewer comments to Author:

Reviewer: 1

Comments to the Author(s)

The authors have made extensive changes to the manuscript and took most concerns previously raised into account. However, I admit I still have substantial reservations about the paper. **The main issue in my view is that a change in integration strength, admitting it is measurable based on behaviour, is not unequivocal proof of changes in causality inference because the weights also depend on the prior which is unknown.** I expressed this concern in my earlier review and I do not feel that the authors have demonstrated that their data could say anything about causality inference. I trust the authors have carefully considered my earlier concern and I am willing to re-consider if there is a flaw in my own reasoning, but as such I cannot endorse the study.

The experiments show that multisensory integration is different if the information is provided before or after the event that they are asked to report. This result is sound and interesting, but any claim about causality inference overstretches the data. **Indeed, it is possible that the results simply reflect a time-dependent integration process, and that when a judgment is made, sensory information provided after has less impact because estimation is not performed backwards. Thus, the prior is given more importance if sensory information is given later. Causality inference in my view is not necessary to explain this data.** In fact, the authors appear to mix the likelihood and a putative inference of causality. That is if the likelihood of feedback is high dependent on their own internal model they will certainly rely strongly on it because the data is correlated with the prediction, but this is not causality inference. If my own reasoning is flawed then perhaps insight from a third angle is needed to arbitrate.

We are not entirely sure what “prior” the reviewer refers to in saying that “the weights also depend on the prior”. The **relative magnitude of the weights** given to the visual and proprioceptive information regarding movement endpoint, can be influenced by several factors including a prior belief regarding the position of the movement endpoint. The **sum of the weights** indicates the integration strength. This integration strength is influenced by the strength of the causality judgment only. The differences in integration strength that we observe in our study are thus a reflection of the underlying differences in causality judgments. We revised the text in the introduction to better explain the direct link between integration strength and causality judgments, and we now state explicitly that we use the integration strength as a measure for the underlying strength of the causality judgment.

Now it does hold that causal inference also relies on a causality prior, a prior that probably builds up over previous experience with computer-use in which the correlation between hand and cursor movement was experienced. We now address this prior more elaborately in the introduction, and we include data from a previous study on the causality prior in the methods section to address the role of this prior in the current experiment (p 20).

Overall the description of the experiment has much improved. There remain problems with the theoretical part:

1a. **I failed to understand the way the model was used** as it seems to me that equations linking λ and the sigmas on page 18 comes from standard maximum likelihood model that does consider a common source for each piece of information. What I failed to understand then was how this was used to assess causality inference since causality was previously assumed to derive these equations.

Standard Maximum likelihood estimation (‘full integration’) indeed assumes a positive causality judgment. What is critical, is that the causality judgment should not be considered an all-or-none estimate, but a probability estimate that reflects the believe that two signals are redundant. The partial integration model that we used here, is an elaboration of the standard Maximum likelihood estimation

model; it can be used to generate model prediction for integration strengths lower than 1, which is what we used it for. For clarification, we revised the text in the Methods section (p 17-18).

1b. In addition, the way predictions were derived was unclear: it is written that the predicted gap was set equal to the observed gap, but in this case the authors performed a post hoc fitting of the data and these are not real model predictions. There is no mechanism to explain how integration strength should evolve prior to seeing the data. I was left with the impression that the model had no explanatory power.

It is important to note that the model comparisons are not the primary outcome in this study; the measure of primary interest is the integration strength. The model is not used to predict the integration strength. We use them to assess whether, for a given integration strength, the biases are consistent with the mechanism of optimal integration. For this reason, the integration strength was the input to the model predictions (this parameter sets the 'gap'), along with the unisensory variabilities. With these predictions we could thus assess whether, for a given integration strength, the weights (i.e., biases) were as one would expect for an underlying reliability-based weighting mechanism. We have rearranged the Results section, to clarify that the model predictions are of much lower importance than the effect of our experimental manipulation on the integration strength. Furthermore, the concerning methods section was adjusted to clarify the minor importance of the model comparisons.

Specific points:

2. First three lines of page 6 are almost impossible to follow.

3. Middle of page 6: the question is also unnecessarily complicated. Why not just recast the paper in terms of asking whether the temporal ordering between sensory information and perception matters?

The temporal ordering of the to-be-judged sensory information and the perceptual judgment is constant in our study. What differs is whether the causality information is provided before or after the to-be-judged sensory information. We have revised the entire introduction section to make the objectives of our study crystal clear.

4. The theoretical framework is missing internal feedback which may or may not be included as a prior, but which is definitely used in the brain.

If with 'internal feedback', the reviewer means the 'efference copy', then we agree with the comment, although one could argue whether this should be called a prior (i.e., based on previous experience) or just a third source of information (i.e., based on current motor outflow). However, in the present experiment and related ones we do not isolate different factors that contribute to the felt position of the hand, e.g. by comparing judgment biases for active and passive movements. Such analyses are clearly beyond the scope of the current study. We agree that they could be interesting – for adaptation to visuomotor rotation corresponding differences have been shown, and they likely exist for (partial) multisensory integration as well (probably with a stronger bias for judgments of endpoints of passive than of active movements).

5. Anovas in the discussion is strange, this could be moved to Results.

This has been moved to the Results now (p. 20).

Reviewer: 2

Comments to the Author(s)

Overall, the authors have done a good job at addressing my comments. I only have a few remaining comments:

1. Looking at Figure 4C there seems to be a clear difference between observed and predicted variability, however this does not reach significance as assessed by the ANOVA.

What we tested was not the effect observed versus predicted (as this comparison was indeed obvious) but whether the difference (i.e., observed minus predicted) differed between the experimental conditions. This section has been changed (e.g., non-parametric test), and should be clear now (p 24).

2. "Figure 3B" in the second paragraph of page 22 should be "Figure 4B"

Has been corrected now.

3. Most of the dataset analyzed here are unlikely to be normal (i.e., standard deviations can only be positive, relative biases and integration strength take value-approximately-between 0 and 1, etc..) however they are analyzed using parametric tests that assume an underlying normal distribution. It would be nice to test for normality and if systematic deviations occur run appropriate transformation or choose different tests (though I guess this might not dramatically change the conclusions).

We have checked for violations of normality, and use non-parametric tests where necessary.

Appendix C

Review of “Exploring the time window for causal inference and the multisensory integration of actions and their visual effects” by Debats and Heuer

Major issue

Lines 30-44 on page 18. I disagree with the authors explanation of a coupling prior. As I understand it, the coupling prior reflects the probability (or expectation) to observe different conflicts between the two cues. A strong coupling prior assumes that, on average, a given world state (in this case the true location of the pen/hand) will give rise to unbiased and equal visual (the cursor on the screen) and proprioceptive (where the observer “feels” their hand to be) estimates, attributing any conflict between estimates from the two inputs to noisy sensory representations and pulling estimates towards the average of the two cues to minimize sensory noise – full integration and cue fusion. A weaker coupling prior allows for the cues to be biased or unequal (the estimate of cursor and hand position from the two sense may be mis-calibrated and not match up perfectly). The weaker the prior, the more conflict is accepted, with the observer maintaining a representation of the two conflicting estimates in case only one is required for a later judgement. If that understanding of the coupling prior is correct then it is, in a sense, a prior that is shaped by experience that captures the redundancy of the two pieces of information. I do agree that it is not a causality prior as there is no option for the cues to come from separate sources. They are assumed to come from the same source but allowed to be mis-calibrated. I think it makes sense to apply such a model to this experiment, as it is true that often the feedback we see when using a mouse, stylus, trackpad, or touch screen can be offset, meaning the observer should maintain access to each individual cue in case they need to make a judgement about each separately. Maybe the authors could do a bit more in this paragraph to explain the relevance of the coupling prior in this experiment and what it captures.

Considering the above, I must agree with reviewer 1, that these results say little about causal inference as the model they compare the data to always assumes the two pieces of information come from a common source. If the authors want to allow for two sources, then they should consider a model comparison between the model detailed here and something akin to the Kording et al. (2007) model that they reference. On page 6 the reviewers seem to suggest that these two models in fact capture the same thing. This may be true (though I don't think it is), but they are at least set up to model two different types of behavior. If they are the same, then the authors need to convince me of this.

Without the causal inference element of the paper, the results are still interesting, possibly revealing the timeframe of integration.

Specific comments

1. Would be helpful to add a sentence to the abstract that makes clear the manipulation (rotation of feedback shown on screen). Without this, it isn't obvious from the abstract why one should expect differences between the conditions.
2. Lines 22-32 on page 3 are sloppy. The first sentence in particular fails to convince me of the link. I suggest a rewrite of these to get the point across better.
3. Lines 29-36 on page 4. The statement that "optimal integration IS the process" is far too strong. Reliability-weighted averaging provides a good prediction of integration behavior, observers often near-optimal in combination of sensory information, but we have no idea if this is what is happening in the brain or if this is achieved via some other heuristic process – one that could explain increasing reports of suboptimalities.
4. Similarly, lines 42-48 on the same page reference four studies then imply reliability-weighted averaging is a "widespread neural phenomenon". This just isn't true. Just as many suboptimalities are reported as near-optimal performance (see Rahnev and Denison, 2018, BBS, Section 3.7 for examples) and I am yet to come across any convincing evidence that neurons are capable of reliability-weighted averaging. I suggest the authors soften this paragraph in general and focus on how reliability-weighted averaging is a good model of behavior (in certain circumstances) but is not necessarily representative of what is going on inside the brain.
5. On page 11 the authors mention a structured post-experimental interview. I am keen for the authors to add more detail about this as I find it striking that observers didn't notice the rotation and think this deserves further discussion.
6. The authors sometimes use the term "sum of the bias" when they really mean "sum of the weights" (e.g. page 20). This should be fixed as they are different quantities.
7. Second paragraph of *Integration Strength* section. The authors imply that the remaining integration strength in the back condition is due to a causality prior, not the causality evidence presented during the back movement as the integration strength in this experiment does not differ to that in another experiment where only the endpoint feedback was shown. I would argue that the endpoint feedback alone provides some causality evidence (feel the barrier at the same time the dot appears so these sensory experiences are correlated) and this analysis may only show that showing the cursor during the back movement adds nothing extra.

Appendix D

Review of “Exploring the time window for causal inference and the multisensory integration of actions and their visual effects” by Debats and Heuer

Major issue

Lines 30-44 on page 18. I disagree with the authors explanation of a coupling prior. As I understand it, the coupling prior reflects the probability (or expectation) to observe different conflicts between the two cues. A strong coupling prior assumes that, on average, a given world state (in this case the true location of the pen/hand) will give rise to unbiased and equal visual (the cursor on the screen) and proprioceptive (where the observer “feels” their hand to be) estimates, attributing any conflict between estimates from the two inputs to noisy sensory representations and pulling estimates towards the average of the two cues to minimize sensory noise – full integration and cue fusion. A weaker coupling prior allows for the cues to be biased or unequal (the estimate of cursor and hand position from the two sense may be mis-calibrated and not match up perfectly). **The weaker the prior, the more conflict is accepted**, with the observer maintaining a representation of the two conflicting estimates in case only one is required for a later judgement. If that understanding of the coupling prior is correct then it is, in a sense, a prior that is shaped by experience that captures the redundancy of the two pieces of information. I do agree that it is not a causality prior as there is no option for the cues to come from separate sources. They are assumed to come from the same source but allowed to be mis-calibrated. I think it makes sense to apply such a model to this experiment, as it is true that often the feedback we see when using a mouse, stylus, trackpad, or touch screen can be offset, meaning the observer should maintain access to each individual cue in case they need to make a judgement about each separately. Maybe the authors could do a bit more in this paragraph to explain the relevance of the coupling prior in this experiment and what it captures.

We apologize for not having been sufficiently clear in explaining the notion of the coupling prior in the model we use. The coupling prior in the model does not indicate the amount of conflict that is accepted, as the reviewer suggests, but it indicates the ‘the systems (un)certainty of two signals belonging together’ (see quote provided below), i.e., an assessment of causality. The certainty represented by the variance of the coupling prior determines the strength of integration, which covers the whole range from full multisensory integration to no integration in the case the signals that are treated as having independent sources. This at least is the conception of the coupling prior held by people who developed the concept (i.e., mainly Ernst), as explained, for example, in a relatively accessible book chapter by van Dam, Parise & Ernst (2016; Modeling multisensory integration). We here quote a paragraph from this book chapter (page 9), in which it is made explicit that the variability of the coupling prior (“coupling uncertainty”) indicates the strength by which the sensory modalities are considered to belong together (i.e., causality estimate in other words), and directly determines the strength of the integration (‘fusion’ here) between the senses. (Note that the parameter σ_{prior}^2 in the manuscript correspond to σ_x^2 in the terminology of the quote.)

In the revised manuscript, we have adjusted the indicated paragraph (p. 17-18), to more clearly explain the meaning of the coupling prior.

Quote: “Depending on the shape of the coupling prior (e.g., a 2-D Gaussian probability distribution) and in particular on the width of the prior along the negative direction, which

represents the coupling uncertainty σ_x , different a posteriori estimates will be obtained. The MLE type of integration in which the senses are completely fused can be illustrated as a prior that enables a one-to-one mapping from one sensory estimate to the other (i.e., an infinitely thin line along the positive diagonal; figure 10.4, bottom row). In this case, **the system is certain the two signals always belong together**, thus the coupling uncertainty σ_x is zero. Here, the combined estimate in the posterior distribution will always end up on the identity line ($SV^{MAP} \square SH^{MAP}$), and information about a discrepancy between the senses would be completely lost. At the other extreme, **if the sensory modalities are considered independent** by the perceptual system, the coupling prior is flat (i.e., the coupling uncertainty σ_x is infinite). In this case, the a-posteriori estimate is fully determined by the likelihood function, and no combined percept is formed (figure 10.4, top row). For intermediate levels of fusion, the coupling prior can be modeled as a bivariate distribution aligned along the diagonal (middle plot). In this case, partial fusion will take place, and there will be perceptual benefit for estimating the property of interest without the information about the separate modalities being completely lost (see figure 10.3). **The narrower the distribution of the coupling prior, the stronger the fusion between the senses.**”

Considering the above, I must agree with reviewer 1, that these results say little about causal inference as the model they compare the data to always assumes the two pieces of information come from a common source. If the authors want to allow for two sources, then they should consider a model comparison between the model detailed here and something akin to the Kording et al. (2007) model that they reference. On page 6 the reviewers seem to suggest that these two models in fact capture the same thing. This may be true (though I don't think it is), but they are at least set up to model two different types of behavior. If they are the same, then the authors need to convince me of this.

As indicate in the reply above, the coupling-prior model does not hold the assumption that the two pieces of information come from a common source. We furthermore want to stress that in this study, the part with the comparison between data and model predictions plays only a minor role. The main analysis focusses around the question: how does the main manipulation (the availability of simultaneous hand and cursor position information during either the outward or backward movement) affect the strength of the integration. The strength of the integration reflects the strength of the underlying causality evidence. This is crucial. If the causality evidence is low (i.e., the probability of separate sources is high) then little integration should result, and vice versa. Given that in the current experiment, the quality of the causality evidence was equivalent in both conditions tested, our findings show that the later-provided causality evidence (condition *Back*) was not as effective as the early-provided causality evidence condition *Out*). This main question regarding the use of early and later causality evidence can, in our view, be answered purely based on the behavioral observations we made here.

The comparisons between data and model predictions that we report were included only to check whether the main manipulation affected the integration in other ways, e.g., in changing the relative weighting of the two sources of position information, and whether the integration again obeys basic principles of multisensory integration as we had found in previous studies with the same type of model. The purpose of the model was not to test specific assumptions about the process of multisensory integration, and any other model incorporating the “reliability principle” of multisensory integration could have served the same purpose – but we kept to the model used in our previous studies for that purpose. We added and adjusted two paragraphs in the discussion (p. 27-29) to clarify the role of the model predictions.

Without the causal inference element of the paper, the results are still interesting, possibly revealing the timeframe of integration.

As emphasized above, we take the strength of integration to reflect the strength of the underlying causality evidence. Hence, the change in integration strength that we observe in our data shows that later-provided causality evidence was less effective than early-provided causality evidence. (Note that we talk about causality evidence in functional terms – as affecting the strength of multisensory integration – and this need not be identical with explicit judgments about the probability of having same or different sources of information; well, the different sources are obvious in our task, so the question to tap conscious awareness of the degree of relatedness or the level of redundancy would have to be phrased differently.)

Specific comments

1. Would be helpful to add a sentence to the abstract that makes clear the manipulation (rotation of feedback shown on screen). Without this, it isn't obvious from the abstract why one should expect differences between the conditions.

Done.

2. Lines 22-32 on page 3 are sloppy. The first sentence in particular fails to convince me of the link. I suggest a rewrite of these to get the point across better.

We expanded the first paragraph to better clarify the link between hand and cursor.

3. Lines 29-36 on page 4. The statement that “optimal integration IS the process” is far too strong. Reliability-weighted averaging provides a good prediction of integration behavior, observers often near-optimal in combination of sensory information, but we have no idea if this is what is happening in the brain or if this is achieved via some other heuristic process – one that could explain increasing reports of suboptimalities.

We changed “is” to “tags”

4. Similarly, lines 42-48 on the same page reference four studies then imply reliability-weighted averaging is a “widespread neural phenomenon”. This just isn't true. Just as many suboptimalities are reported as near-optimal performance (see Rahnev and Denison, 2018, BBS, Section 3.7 for examples) and I am yet to come across any convincing evidence that neurons are capable of reliability-weighted averaging. I suggest the authors soften this paragraph in general and focus on how reliability-weighted averaging is a good model of behavior (in certain circumstances) but is not necessarily representative of what is going on inside the brain.

We agree and omitted the sentence

5. On page 11 the authors mention a structured post-experimental interview. I am keen for the authors to add more detail about this as I find it striking that observers didn't notice the rotation and think this deserves further discussion.

This not-perceiving of discrepancies is indeed a striking aspect of human perception, that has been reported before (cf. Fournieret and Jeannerod 1998; Sülzenbrück and Heuer 2009; Müsseler and Sutter 2009). In a previous study with a similar paradigm, we explicitly informed participants about the rotation. Even then, the majority of the participants could not perceive it. This manipulation did, however, lead to a small reduction in integration strength. We now refer this study (Debats & Heuer, 2018, explicit knowledge of sensory non-redundancy can reduce the strength of multisensory integration) and the three others, in the revised manuscript.

6. The authors sometimes use the term “sum of the bias” when they really mean “sum of the weights” (e.g. page 20). This should be fixed as they are different quantities.

We agree and corrected the phrasing

7. Second paragraph of *Integration Strength* section. The authors imply that the remaining integration strength in the back condition is due to a causality prior, not the causality evidence presented during the back movement as the integration strength in this experiment does not differ to that in another experiment where only the endpoint feedback was shown. I would argue that the endpoint feedback alone provides some causality evidence (feel the barrier at the same time the dot appears so these sensory experiences are correlated) and this analysis may only show that showing the cursor during the back movement adds nothing extra.

We have adjusted the phrasing into (p. 20): “This remaining integration strength could be based on the causality evidence provided during the backward movement, and/or on other sources of causality information. These sources could be the causality prior, and possibly additional causality information being present when the hand and cursor are both in the endpoint, at different endpoints across trials.”

Appendix E

I'm sorry, but I still cannot agree with the authors on the definition of a coupling prior. In "A Bayesian view on multimodel cue integration", part of "Human Body Perception From The Inside Out" (2006), Ernst writes:

"How can the different levels of interaction that determine the strength of coupling between the two sets of cues be understood? As described above, in case of complete fusion the system loses access to the individual estimates. If for some reason, however, the system needs to retain access to the individual estimates, it makes sense that the system does not fuse the signals completely. One obvious candidate for when it is necessary to retain access to the individual estimates is if the mapping between signals is not fixed, but changes in response to a constant conflict between signals. Without some degree of access to the individual estimates it would be impossible to detect conflicts between signals and so it would be impossible for the system to change the mapping between signals in order to overcome the conflict."

This clearly refers to the **conflict** between the signals. Later, Ernst goes on to say:

"The mapping between the different signals can fluctuate on different time-scales. For adapting the mapping between signals quickly, a reliable estimate of the conflict is needed; for adapting slowly, each estimate of the conflict does not need to be very precise – only the average over many observations must yield a reliable conflict estimate. Strong coupling between signals that introduces a strong perceptual bias yields a less reliable estimate of possible conflicts than does weak coupling between signals. If the relationship between signals derived from the same object or event is never changing (the mapping is constant), the system does not need to retain access to the individual estimates and the signals can be completely fused."

Here Ernst states that the coupling prior is considering "**signals derived from the same object or event**" – a common source. This is followed by a more concrete definition of the coupling prior:

"[T]he "Coupling Prior" ... relates to the probability for knowing the mapping between sensory signals. If the mapping is known for sure, signals can be fused; contrary, if the mapping is unknown, signals should be kept separate."

Putting all this together, it seems to me that the "mapping" describes the conflict between the two signals, or how well aligned they are. Ernst then goes on to discuss this with an example very relevant for this study – tool use.

"Why does the mapping between signals change? In order to be robust against changes in the environment or changes occurring to the body our perceptual and perceptual-motor system has to

be very flexible. The system can compensate for such changes through the process of adaptation, which corresponds to a change in mapping between signals. For example, human beings are very skilled at using tools. Using a tool often requires a mapping from visual coordinates to the coordinates of the tools end-effector. Thus, using tools can be seen as an extension of the body that requires adaptation (see chapter by Làdavas & Famè in this volume). Fusing signals mandatory so that the system loses access to the incoming information would prevent such adaptation from happening, because the discrepancy between the signals could not be detected. Without the discrepancy being detected, the error signal necessary for adaptation (the remapping of signals) is missing. A reliable error signal allows for quick adaptation; adaptation to an unreliable error signal should be slow. Therefore, whenever quick adaptation is necessary (e.g., for remapping of vision to body sense during tool use – plasticity of peripersonal space) signals should not be tightly coupled maintaining the chance to reliably detect an error signal; contrary, when adaptation can be on a longer time frame (as argued for the disparity-texture example from Exp. 2) coupling can be stronger and so the signals are fused more completely.”

I don't see anything in the quote provided by the authors that would suggest Ernst's thinking on the matter has changed from 2006 to 2016. Possibly the authors are hanging on to the use of the word “independent” as evidence that Ernst considers that the two signals could come from separate sources. I would argue though that the two signals could be independent (i.e. they provide differing information and are not redundant when considered alongside each other) but still come from a common source. This is my interpretation of Ernst's coupling prior.

It may seem like we are just discussing semantics at this point, but I feel there is an important point to make here. Ernst seemed to develop the coupling prior to capture instances of partial integration that could in fact be considered *optimal*. It isn't designed to model situations where an observer separates out incoming sensory information, inferring that different signals come from different sources (that is what the Kording et. al.'s (2007) model is designed to do), but rather, it is designed to capture the fact that incoming sensory signals from the same source may sometimes be mismatched. In that case, it is beneficial to keep access to each individual piece of information and avoid complete fusion.

Now I agree that one could use Ernst's coupling prior to model inferences of multiple sources too, but the issue is that one cannot tell apart the cause of partial integration (inference of two causes or inference of weak coupling/large conflict). That is why I agreed with reviewer 1, though realise I failed to make this point well in the initial review. If the authors want to untangle this then they need some form of model comparison. Otherwise, all they can say is that there is a difference

between Out and Back conditions but cannot say this is because observers are less likely to infer a common cause in the latter. It could be that observers perceive a greater mis-match/conflict between the proprioceptive and visual information in the Back compared to the Out condition as the two signals will be further separated in the former condition compared to the latter (when the visual signal first appears) due to the radial nature of the task. In the Out condition, the signals go from low to high conflict but the opposite is true in the Back condition.

I'll stress again that I still think the results here are interesting, but the authors need to relax their causal inference claims.

Appendix F

Dear Editors,

Reviewer 2 is concerned that our findings might be caused by a different process than causal inference. The reviewer is basically suggesting that there might exist two different processes that determine the degree to which two (or, multiple) signals are to be integrated: 1) ‘causal inference’ would be the process that estimate a probability that signal arise from a common source (object/event) and thereby setting the degree of integration. 2) Another – unnamed – process would estimate the necessity to keep information from a single source apart for the purpose of potentially needed adaptation / recalibration.

We have not before come across this notion of a dedicated process that determines the degree-to-integrate given the necessity-to-recalibrate. We agree that the relationship between integration and recalibration is highly interesting and thought-provoking. On the one hand there is a prominent view that integration is a precursor of adaptation (e.g., Hay et al., 1965; Wozny & Shams, 2011). On the other hand, integration and adaptation might be independent processes, as integration does not necessarily lead to recalibration (e.g., Smeets et al., 2006). Along another line of thought, it seems that the stronger one integrates two pieces of information, the smaller the conflict that can be perceived and thus the smaller the ‘error-signal’ that can drive adaptation/recalibration. Ernst also describes this in the 2006 book chapter, from which the reviewer quotes. He does, however, not propose the existence of a separate mechanism for ‘the degree-to-integrate given the necessity-to-recalibrate’, nor does he propose the coupling prior model to be a model dedicated to adaptation/recalibration. This can also be appreciated from a 2014 review paper, in which adaptation/recalibration is only addressed in the very last section (van Dam, Parise & Ernst (2014), Modeling multisensory integration). Furthermore, we know of no other scientific publications that supports the existence of a dedicated process, besides causal inference, that detemines the strength of multisensory integration.

The reviewer comments that we cannot untangle which of the two processes (causal inference or the novel process as proposed by the reviewer) determined the degree of integration in our task. Based on that, he/she suggests that we should either include model comparisons to substantiate our statements on causal inference, or we should relax our causal inference claim.

Neither of the two adjustments to the manuscript is appropriate, we believe, first and foremost, because of the absence of evidence for another process besides causal inference that determines the strength of integration. Furthermore, model comparisons would not succeed in separating the role of causal inference from the role of a different process – the hypothesized causal inference would only be mapped on different parameters such as the probabilities of full vs no integration or the variance of the “coupling prior”. Additionally, as we previously replied to another reviewer, model comparisons go beyond the scope of this work. What our data show is a clear difference in integration strength that suggests that the availability of correlated hand-cursor movement (the causality evidence here) is much more effective before to-be-integrated sensory information (movement endpoints here) is provided, compared to when it is provided directly thereafter. This is our key finding and it stands independently of what model is used to further describe/capture the data. This minor role of the model predictions is stressed at several places in the manuscript, including in the results section of the revised manuscript (p. 23).

Therefore, we resubmit the manuscript with minor textual improvements. We furthermore hope that our ongoing work, which addresses the relation the between integration and adaptation, will get out in the near future to feed the debate on the integration-recalibration topic that seems of interest to the reviewer.